# The transcription factors TFE3 and TFEB amplify p53 dependent transcriptional programs in response to DNA damage

Owen A Brady[1†], Eutteum Jeong[1†], José A Martina[1], Mehdi Pirooznia[2], Ilker Tunc[2], Rosa Puertollano[1*]

[1]Cell and Developmental Biology Center, National Heart, Lung, and Blood Institute, National Institutes of Health, Maryland, United States; [2]Bioinformatics and Computational Biology Core, National Heart, Lung, and Blood Institute, National Institutes of Health, Maryland, United States

**Abstract** The transcription factors TFE3 and TFEB cooperate to regulate autophagy induction and lysosome biogenesis in response to starvation. Here we demonstrate that DNA damage activates TFE3 and TFEB in a p53 and mTORC1 dependent manner. RNA-Seq analysis of TFEB/TFE3 double-knockout cells exposed to etoposide reveals a profound dysregulation of the DNA damage response, including upstream regulators and downstream p53 targets. TFE3 and TFEB contribute to sustain p53-dependent response by stabilizing p53 protein levels. In TFEB/TFE3 DKOs, p53 half-life is significantly decreased due to elevated Mdm2 levels. Transcriptional profiles of genes involved in lysosome membrane permeabilization and cell death pathways are dysregulated in TFEB/TFE3-depleted cells. Consequently, prolonged DNA damage results in impaired LMP and apoptosis induction. Finally, expression of multiple genes implicated in cell cycle control is altered in TFEB/TFE3 DKOs, revealing a previously unrecognized role of TFEB and TFE3 in the regulation of cell cycle checkpoints in response to stress.
DOI: https://doi.org/10.7554/eLife.40856.001

*For correspondence:
puertolr@mail.nih.gov

†These authors contributed equally to this work

Competing interests: The authors declare that no competing interests exist.

## Introduction

TFEB and TFE3 belong to the MiTF-TFE family of basic helix-loop-helix leucine zipper (bHLH-LZ) transcription factors and have been well characterized as master regulators of the autophagy-lysosome pathway in response to starvation (*Martina et al., 2014*; *Sardiello et al., 2009*; *Settembre et al., 2011*). Under nutrient poor conditions, TFEB and TFE3 translocate to the nucleus where they bind to and promote transcription from consensus 10 bp GTCACGTGAC sequences, termed Coordinated Lysosome Expression and Regulation (CLEAR) elements, in the promoters of autophagic and lysosomal genes, (*Martina et al., 2014*; *Palmieri et al., 2011*; *Sardiello et al., 2009*; *Settembre et al., 2011*). Under conditions of adequate nutrient status, TFEB and TFE3 bind to a pair of heterodimeric small GTPases, termed RagA or RagB (RagA/B) and RagC or RagD (RagC/D) on the lysosome surface (*Martina and Puertollano, 2013*). These Rag GTPases also recruit the mechanistic target of rapamycin (mTOR) complex 1 (mTORC1), which phosphorylates TFEB at Ser211 and TFE3 at Ser321, thus creating a binding site for proteins in the 14-3-3 family which sequester TFEB and TFE3 in the cytosol by masking their nuclear localization signals (*Martina et al., 2012*; *Martina et al., 2014*; *Powis and De Virgilio, 2016*; *Roczniak-Ferguson et al., 2012*).

Since all cells require the homeostatic functions of the autophagy-lysosome pathway, these nutrient responsive mechanisms of control over TFEB and TFE3 are highly conserved. Recently, other cellular stressors such as inflammatory or pathogenic signals, ER stress, oxidative stress, and high intensity exercise have been implicated in activating TFEB and TFE3 via a variety of mechanisms

involving mTORC1 as well as other kinases and phosphatases, including calcineurin and PP2A (*Chen et al., 2017*; *Mansueto et al., 2017*; *Martina et al., 2016*; *Martina and Puertollano, 2018*; *Najibi et al., 2016*; *Pastore et al., 2016*; *Visvikis et al., 2014*). In addition to being regulated by multiple physiological stressors, TFEB and TFE3 exhibit unique cell or tissue type specific responses including pro-inflammatory, chemoattractant, and antimicrobial responses in immune cells such as macrophages and dendritic cells; lipid metabolism in liver and adipose tissue; and mitochondrial bio-genesis in muscle (*Bretou et al., 2017*; *Mansueto et al., 2017*; *Najibi et al., 2016*; *Pastore et al., 2016*; *Settembre et al., 2013*; *Visvikis et al., 2014*).

The tumor suppressor transcription factor p53 integrates DNA damage signals resulting in tran-scriptional programs governing the diverse processes of the DNA damage response (DDR) including cell cycle arrest, DNA repair, senescence, and when the stress is severe enough, apoptosis (*Bieging and Attardi, 2012*). Much like TFE3 and TFEB, p53 has been shown to respond to multiple cellular stressors including nutrient deprivation, oxidative stress, and hypoxia. Similarly, p53 has roles controlling transcriptional activity towards processes not canonically associated with the DDR, includ-ing autophagy, cell migration, and metabolism (*Bieging and Attardi, 2012*; *Nishida et al., 2016*). p53 activity is controlled through an extensive repertoire of posttranslational modifications at approximately 50 residues which include phosphorylation, acetylation, methylation, ubiquitylation, sumoylation, and poly-ADP-ribosylation (*Meek and Anderson, 2009*). Under baseline conditions, p53 is rapidly ubiquitylated and degraded by the E3 ligase, Mdm2, limiting its activity. In response to DNA damage and other stressors, p53 is rapidly phosphorylated and acetylated by a large variety of kinases and acetyltransferases which serve to stabilize and modulate binding to p53 response ele-ments in the promoters of different classes of genes (*Meek and Anderson, 2009*).

Importantly, p53 has critical and multifaceted roles in regulating autophagy, including a suppres-sive function attributed to its cytosolic form and a more well-established pro-autophagic function owing to its transcriptional activity in the nucleus (*Kenzelmann Broz et al., 2013*; *Tasdemir et al., 2008*). Autophagy itself appears to be induced as part of the DDR. For example, an alternative Atg5 and LC3-independent form of autophagy is induced in mouse embryonic fibroblasts in response to the DNA damaging drug, etoposide (*Nishida et al., 2009*). Autophagic functions critical for the DDR include ensuring an adequate supply of NAD$^+$ and dNTPs for repair functions, maintaining genomic stability through removal of micronuclei containing chromosome fragments, and the assem-bly of the nucleotide excision repair (NER) complex at sites of DNA damage (*Eliopoulos et al., 2016*). Finally, many gene products, including the p53 regulated Dram1 and Trp53inp1 are involved in autophagic processes while exhibiting independent or complementary functions in the DDR.

The role of p53 in cancer has been studied extensively, with approximately half of all human can-cers exhibiting mutations at the *TP53* locus, while cancers without p53 mutations frequently have other alterations in the p53 pathway (*Eliopoulos et al., 2016*). While not as widely associated with all cancers, TFE3 and TFEB gene fusions are detected in subsets of renal cell carcinomas (RCC), indi-cating roles for these transcription factors in oncogenesis (*Kauffman et al., 2014*). Given the com-monalities between TFE3/TFEB and p53 in regard to their activation by diverse cellular stressors and their shared roles in the transcriptional control of autophagy and other cellular stress responses, we wondered if TFE3 and TFEB exhibited any common regulatory mechanisms with p53. In this study, we report that TFE3 and TFEB are indeed activated by DNA damage, albeit with a delayed kinetic profile compared to p53. This response is at least partially dependent upon p53-mediated mTORC1 inhibition. RNA-Seq analysis of MEFs and RAW264.7 cells treated with etoposide reveals a robust DDR with upregulation of canonical p53 regulated transcripts, which is strongly dysregulated in a CRISPR/Cas9 generated TFE3/TFEB double knockout (DKO) background. Conversely, overexpres-sion of constitutively active TFE3 and TFEB mutants increases expression of DDR genes involved in upstream transduction of DDR signals along with downstream DDR effectors, including genes involved in apoptosis and p53 itself. Lastly, we relate defects in TFE3 and TFEB signaling in response to DNA damage to defects in lysosome membrane permeabilization (LMP), regulation of cell death pathways, and cell cycle control. These data are the first reported example of extensive cross-talk between the p53 and TFE3/TFEB pathways showing disruption in expression of DDR genes as well as functional defects in lysosome integrity and cell death pathways in TFE3/TFEB-depleted cells.

## Results

### TFE3 and TFEB translocate to the nucleus in response to genotoxic stress

We and others have previously demonstrated that TFE3 and TFEB are activated in a variety of cell types in response to various physiological stressors. The best described pathway is in response to starvation and is contingent upon mTORC1 inhibition (*Martina et al., 2012*; *Martina et al., 2014*; *Roczniak-Ferguson et al., 2012*; *Settembre et al., 2012*). More recently, we have shown that TFE3 and TFEB are activated in response to ER stress and in response to pathogenic signatures in macrophages, both in an mTORC1 independent fashion (*Martina et al., 2016*; *Pastore et al., 2016*). To determine if genotoxic stress activates TFE3 and TFEB, we exposed wild type mouse embryonic fibroblasts (MEFs) to etoposide, a drug which induces both single strand breaks and double strand breaks in DNA which are corrected via a variety of repair pathways (*de Campos-Nebel et al., 2010*; *Pommier et al., 2010*). We observed a time-dependent increase in the levels of TFE3 in the nucleus relative to the cytosol, with noticeable accumulation appearing by 4 hr treatment, and increasing amounts at 8 hr and 16 hr (*Figure 1A*). After 8 hr treatment, the nuclear localization of TFE3 is statistically significant compared to control (*Figure 1B*; numerical values *Figure 1—source data 1*). While the proportion of live cells exhibiting nuclear TFE3 increases further by 16 hr treatment, the number of dead cells begin to outnumber the living, hence 8 hr was used as a benchmark for a number of subsequent experiments with etoposide as a compromise between maximal TFE3 activation and minimal cell death.

To rule out the possibility that etoposide exerted its effect on TFE3 translocation through a mechanism independent of its DNA damaging effects, we tested additional DNA damaging agents. Unlike etoposide, the chemotherapeutic agent cisplatin damages DNA by forming guanine adducts and intrastrand cross-links, which are preferentially repaired by the nucleotide excision repair (NER) pathway (*O'Grady et al., 2014*). Despite this mechanistic difference in mode of DNA damage, cisplatin treatment also effectively induced TFE3 nuclear translocation in MEFs (*Figure 1C*). Treatment with ultraviolet light, which primarily damages DNA through the formation of pyrimidine dimers (*Gentile, 2003*), also induced TFE3 activation in MEFs (*Figure 1C*). We also confirmed that TFE3 nuclear accumulation in response to genotoxic stress was observed in other cell types, including ARPE-19, HeLa and RAW 264.7 (*Figure 1—figure supplement 1A–C*). Finally, we performed subcellular fractionation experiments following treatment with either etoposide or cisplatin. As seen in *Figure 1D* and *Figure 1—figure supplement 1H*, we observed a significant increase in the amount of nuclear TFE3 and TFEB following DNA damage.

Activation of TFE3 requires dephosphorylation of serine 321. Treatment of several cell types with different DNA damaging agents, resulted in a time dependent decrease in S321 phosphorylation that correlated with the degree of TFE3 nuclear translocation observed by immunofluorescence (*Figure 1E* and *Figure 1—figure supplement 1D–G*). Starvation in Earle's Balanced Salt Solution for 2 hr was used as a positive control for dephosphorylation (*Figure 1E*). TFEB serine 211 (S211) is the equivalent residue to TFE3 S321, however phospho-specific antibodies generated against this residue are not detectable with endogenous TFEB levels. Nonetheless, TFEB exhibits a pronounced gel-shift following DNA damage, which is indicative of dephosphorylation, thus suggesting a similar regulation of both transcription factors in response to DNA damage (*Figure 1E* and *Figure 1—figure supplement 1D–G*). Furthermore, treatment of a HeLa clone that stably expresses TFEB-FLAG with UV-C light resulted in a rapid dephosphorylation of TFE3 and TFEB at S321 and S211, respectively (*Figure 1F*).

Taken together, these data indicate that TFE3 and TFEB are activated by a wide variety of DNA-damaging stimuli with varying mechanisms of action and preferential DNA repair pathway responses.

### Activation of TFE3 and TFEB in response to genotoxic stress is dependent on mTORC1 and p53

Activation of TFE3 and TFEB in response to starvation is dependent on mTORC1 inactivation and subsequent dephosphorylation of TFE3-S321 and TFEB-S211. Other stressors, such as ER stress, pathogen infection and oxidative stress, result in TFE3 and TFEB activation despite high mTORC1

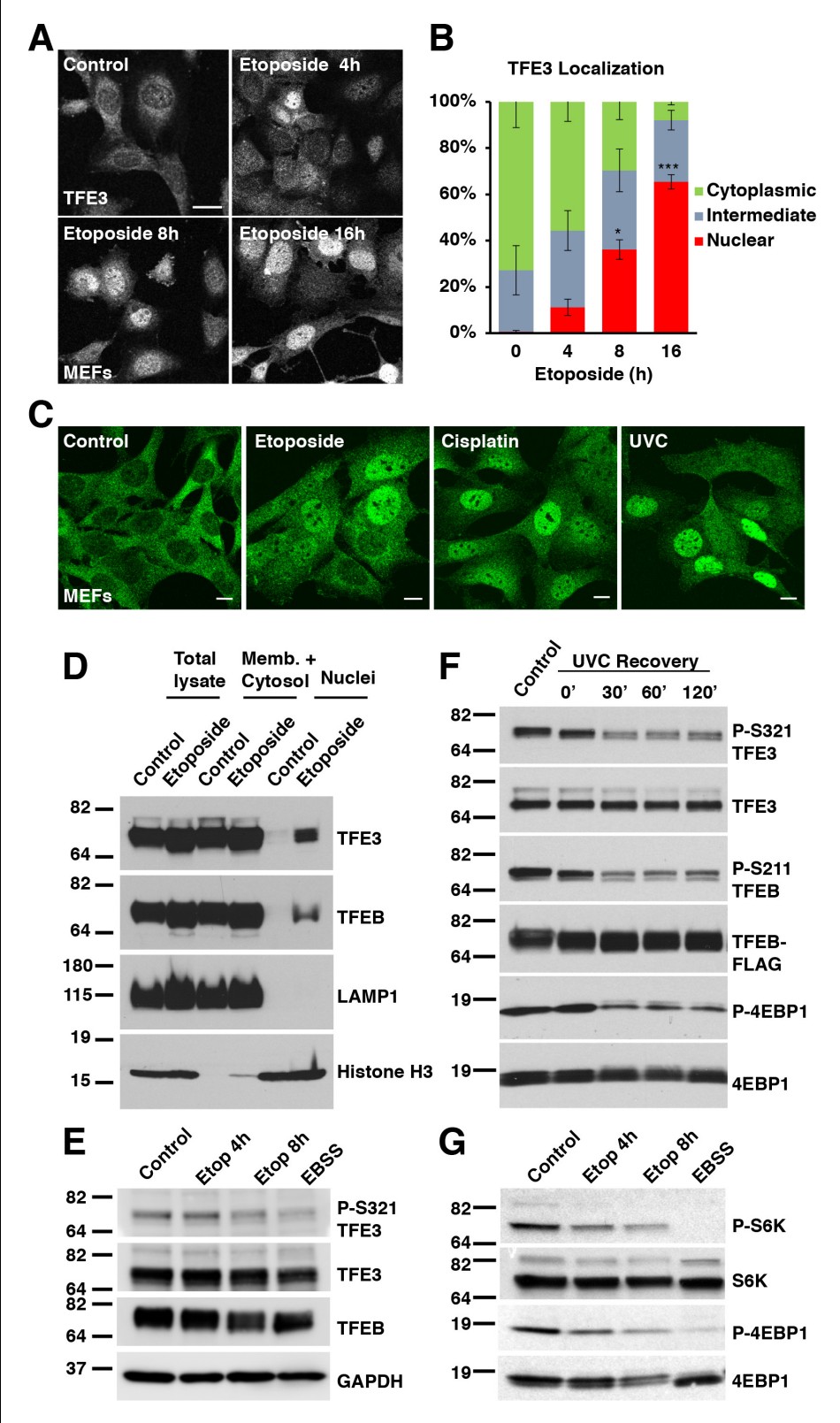

**Figure 1.** TFE3 and TFEB translocate to the nucleus in response to genotoxic stress. (**A**) Immunofluorescence images of WT MEFs treated with 100 μM etoposide for up to 16 hr. Scale bar = 20 μm. (**B**) Quantification of TFE3 localization from cells shown in A. Levels represent mean ± standard deviation with n = 3 experiments and > 200 cells counted per trial. Significance determined with Student's t-test (*p < 0.05, ***p < 0.001). (**C**)

*Figure 1 continued on next page*

*Figure 1 continued*

Immunofluorescence images displaying TFE3 translocation in WT MEFs in response to different DNA damaging agents: etoposide, cisplatin, and UVC irradiation, and Scale bar = 10 µm. (D) Representative Western blot showing TFE3 and TFEB nuclear distribution by subcellular fractionation of WT MEFs exposed to etoposide for 10 hr. (E) Representative Western blot showing TFE3 de-phosphorylation at Ser321 and gel shift in TFEB in WT MEFs exposed to etoposide for up to 8 hr. EBSS for 2 h hours used as a positive control for de-phosphorylation of TFE3 and TFEB. (F) Representative Western blot showing TFE3 and TFEB de-phosphorylation in HeLa cells in response to UV-C light. (G) Representative Western blot showing mTORC1 inhibition due to S6K and 4EBP1 de-phosphorylation in response to increasing etoposide treatment time. EBSS for 2 hr used as a positive control for maximum mTORC1 inhibition. All the western blots are representative of three independent experiments.

DOI: https://doi.org/10.7554/eLife.40856.002

The following source data and figure supplement are available for figure 1:

**Source data 1.** TFE3 localization with etoposide treatment.

DOI: https://doi.org/10.7554/eLife.40856.003

**Figure supplement 1.** (A) Immunofluorescence images of ARPE-19 cells treated with 100 µM etoposide or 50 µM Cisplatin for 24 hr or 10 hr after UVC irradiation.

DOI: https://doi.org/10.7554/eLife.40856.004

---

activity. This mTORC1-independent activation is likely due to activation of specific phosphatases, such as calcineurin and PP2A (*Martina and Puertollano, 2018*; *Martina et al., 2016*; *Medina et al., 2015*; *Pastore et al., 2016*). To test if mTORC1 activity is affected under DNA damage conditions, we performed western bloting on lysates from cells treated with etoposide or starved with EBSS as a positive control. As expected, EBSS treatment resulted in a near complete inhibition of mTORC1 activity as assessed by phosphorylation of S6K and 4EBP1 (*Figure 1G*). Treatment with etoposide also resulted in a time-dependent decrease in S6K and 4EBP1 phosphorylation levels that mirrors the kinetics of observed TFE3 nuclear translocation, suggesting that mTORC1 inhibition may be a major mechanism by which DNA damage promotes TFE3 and TFEB activation (*Figure 1G*). It is notable that the degree of mTORC1 inhibition after 8 hr etoposide was not as potent as that induced by starvation and closely correlates with the degree of dephosphorylation and nuclear redistribution of TFE3 under etoposide treated and starvation conditions, respectively. The effect of DNA damage on reducing mTORC1 activity was also observed in ARPE-19 cells, which could tolerate extended etoposide treatment past 24 hr (*Figure 1—figure supplement 1I*), as well as in HeLa cells treated with UVC (*Figure 1F*).

Several studies have reported that the activation of the DNA damage response (DDR) leads to p53-dependent mTORC1 inactivation (*Budanov and Karin, 2008*; *Cam et al., 2014*; *Feng et al., 2005*). Given these connections between DNA damage, p53 activity, and mTORC1 inactivation, we hypothesized that the TFE3 and TFEB activation observed in our experiments was dependent on p53 activation. We tested this by repeating mTORC1 activity assays and TFE3 translocation assays in both wild type (WT) and p53[-/-] MEFs. As predicted, etoposide treatment resulted in a time-dependent decrease in mTORC1 activity in WT MEFs, whereas this decrease was virtually absent in p53[-/-] MEFs (*Figure 2A*). Unlike with DNA damage, p53[-/-] MEFs still displayed a strong reduction in mTORC1 signaling in response to starvation, although the levels remained slightly elevated when compared with WT MEFs (*Figure 2A*). Quantification of mTORC1 activity in these two cell lines reveals significant reductions in S6K and 4EBP1 phosphorylation starting at 4 hr etoposide treatment (*Figure 2B*; numerical values *Figure 2—source data 1*). TFE3 localization, as assessed by immunofluorescence, revealed a significantly reduced proportion of cells with TFE3 in the nucleus in p53[-/-] compared to WT MEFs (*Figure 2C and D*; numerical values *Figure 2—source data 2*). Similar results were obtained when analyzing TFEB distribution, with a significant reduction in the number of cells showing TFEB nuclear accumulation in etoposide-treated p53[-/-] MEFs (*Figure 2E and F*; numerical values *Figure 2—source data 3*). Moreover, western blots of lysates from WT MEFs treated with etoposide revealed prominent gel shifts in TFE3 and TFEB, which were absent in etoposide treated p53[-/-] MEFs (*Figure 2—figure supplement 1A*).

To further confirm the role of mTORC1 in TFEB/TFE3 activation in response to genotoxic stress, we assessed whether constitutive mTORC1 activation prevents nuclear translocation of these transcription factors under DNA damage conditions. For this, we transfected ARPE-19 with a constitutive

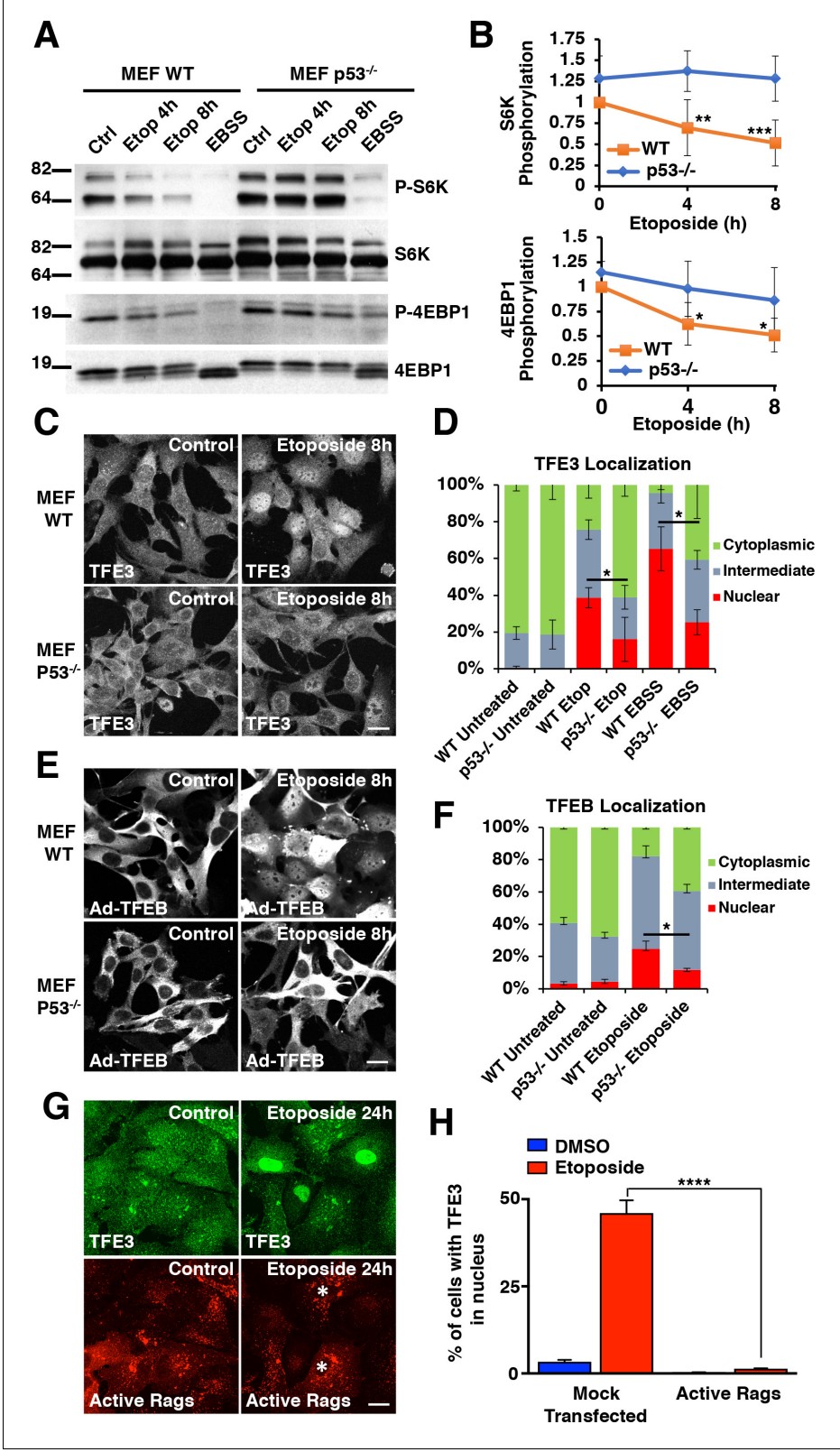

**Figure 2.** DNA damage-induced TFE3 and TFEB activation is a p53 and mTORC1 dependent process. (A) Representative Western blot showing p53-dependent inhibition of mTORC1 in response to etoposide treatment in WT and p53[-/-] MEFs. EBSS for 2 hr was used as a positive control for mTORC1 inhibition and was relatively unaffected by p53 status. (B) Quantification of Western blot data shown in A. Values represent mean ± standard

*Figure 2 continued on next page*

*Figure 2 continued*

deviation with n = 5. Significance determined with Two-way ANOVA with Sidak's multiple comparisons test (*p < 0.05, **p < 0.01, ***p < 0.001). (C) Immunofluorescence images displaying TFE3 translocation after 8 hr etoposide in WT MEFs compared to p53$^{-/-}$ MEFs. Scale bar = 20 μm. (D) Quantification of results from C. Levels represent mean percentage of cells localized in the nucleus, cytosol, or evenly distributed between both with n = 3 and > 80 cells counted per condition per trial. Significance determined with Student's t-test (*p < 0.05). (E) Immunofluorescence images displaying TFEB translocation after 8 hr etoposide in WT and p53$^{-/-}$ MEFs infected with Ad-TFEB-FLAG. Scale bar = 20 μm. (F) Quantification of results from E. Levels represent mean percentage of cells localized in the nucleus, cytosol, or evenly distributed between both with n = 2 and > 80 cells counted per condition per trial. Significance determined with Student's t-test (*p < 0.05). (G) Immunofluorescence images displaying TFE3 cellular distribution after 24 hr etoposide in ARPE-19 cells expressing active Rag heterodimers. Asterisks indicate transfected cells. Scale bar = 10 μm. (H) Quantification of results from G. Values represent mean ± standard deviation of the percentage of cells with nuclear TFE3 with n = 2 experiments and > 300 cells counted per trial (****p < 0.0001).

DOI: https://doi.org/10.7554/eLife.40856.005

The following source data and figure supplements are available for figure 2:

**Source data 1.** Phosphorilation of S6K and 4EBP1 in WT and p53-/- MEFs.
DOI: https://doi.org/10.7554/eLife.40856.006

**Source data 2.** TFE3 localization in WT and p53-/- MEFs.
DOI: https://doi.org/10.7554/eLife.40856.007

**Source data 3.** TFEB localization in WT and p53-/- MEFs.
DOI: https://doi.org/10.7554/eLife.40856.008

**Source data 4.** TFE3 nuclear translocation in response to etoposide in cells transfected with active Rags.
DOI: https://doi.org/10.7554/eLife.40856.009

**Figure supplement 1.** (A) Representative Western blot showing TFEB and TFE3 gel shifts in response to etoposide in WT MEF, but not in p53$^{-/-}$ MEF.
DOI: https://doi.org/10.7554/eLife.40856.010

**Figure supplement 1—source data 1.** qPCR data for lysosomal-autophagy genes.
DOI: https://doi.org/10.7554/eLife.40856.011

active version of Rag GTPases, in which RagB and RagD are locked in their GTP-bound and GDP-bound state, respectively, thus becoming insensitive to mTORC1 inactivation by the p53-sestrin-GATOR1 axis. As expected, expression of active Rags, and consequent constitutive mTORC1 activation, prevented translocation of endogenous TFE3 to the nucleus following etoposide treatment (*Figure 2G and H*; numerical values *Figure 2—source data 4*). These data corroborate that mTORC1 inactivation is required for TFE3 activation in response to DNA damage.

We also observed a statistically significant reduction in the nuclear TFE3 population in p53$^{-/-}$ MEFs exposed to starvation conditions, suggesting that p53 may play an unexpected role in the regulation of the starvation response with regards to regulation of TFE3 and TFEB (*Figure 2D*). Accordingly, transcriptional upregulation of several TFEB and TFE3 targets, including Lamp1, Mcoln1, Atp6vc1, and Ctsd, was significantly reduced in p53$^{-/-}$ MEFs (*Figure 2—figure supplement 1B*; numerical values *Figure 2—figure supplement 1—source data 1*). Despite its best characterized role as a master transcriptional regulator of the DDR, p53 has a well-established, but complex roles in the regulation of autophagy. Activated p53 is involved in upregulating autophagy in many experimental contexts but has also been implicated in suppressing basal autophagy under other experimental conditions (*Feng et al., 2005*; *Tasdemir et al., 2008*; *Zeng et al., 2007*). Given the p53 dependence of TFE3 and TFEB activation in response to DNA damage and the fact that these transcription factors are themselves critical transcriptional regulators of the autophagy and lysosomal pathways, an attractive hypothesis is that p53's effects on these pathways are at least partially dependent on TFE3 and TFEB activity. Nonetheless, direct p53 binding to p53 response elements (PRE) in the promoters of autophagic and lysosomal genes such as cathepsin D have been reported, suggesting a more complex interplay between these transcription factors in these pathways (*Wu et al., 1998*).

## Alterations in the DNA Damage Response in TFEB/TFE3 knockout cells

In order to obtain a global view of the effects of TFE3 and TFEB on the DDR transcriptome, we performed RNA-Seq analysis of WT and TFEB/TFE3 DKO MEFs under basal conditions and in response to 8 hr etoposide treatment. Treatment of WT MEFs induced a potent upregulation of typical p53 regulated DDR genes involved in cell cycle arrest, autophagy, and apoptosis. These included Cdkn1a, Mdm2, Trp53inp1, and many others, confirming that our control cell line behaves in an appropriate manner in response to DNA damage insults (*Figure 3*; numerical values *Figure 3— source data 1*).

In contrast to the WT MEFs, TFEB/TFE3 DKO MEFs exhibited a large-scale dysregulation of the DDR, with numerous p53-regulated and p53 regulating transcripts significantly downregulated after exposure to etoposide, including, Rad9a, Chek2, Bbc3, Bax, Trp53inp1, Dram1, Mdm2, Sesn1, and Sesn2, among others (*Figure 3—figure supplement 1*). Expression levels of these genes were verified using qRT-PCR (*Figure 3*). The proximal DDR response genes Rad9a and Chek2 are both involved in p53 activation, either directly, as with the p53 kinase activity of Chek2, or indirectly as in the case of Rad9a in its capacity as a DNA damage sensor as part of the 9-1-1 complex (*Lim et al., 2015*; *Zannini et al., 2014*). Additionally, Rad9a has been shown to directly interact with and assist in p53 transactivation of certain genes such as Cdkn1a (*Ishikawa et al., 2007*). The levels of both Rad9a and Chek2 were strongly and significantly downregulated in TFEB/TFE3 DKO MEFs in response to etoposide treatment.

The remaining genes in this list represent well-characterized, direct transcriptional targets of p53. Of these genes, Trp53inp1 and Mdm2, exhibit direct p53 regulatory roles on p53 itself. Trp53inp1 encodes a multifunctional protein involved in antioxidant response, autophagy, and cell death pathways (*Cano et al., 2009*; *Okamura et al., 2001*; *Seillier et al., 2012*). Importantly, Trp53inp1 also exhibits positive feedback towards p53 pro-apoptotic function by promoting phosphorylation of p53 at Ser46 by the kinase HIPK2 (*Okamura et al., 2001*; *Tomasini et al., 2003*). Mdm2 encodes an E3 ubiquitin ligase that promotes the degradation of p53 and as a direct transcriptional target of p53 exerts feedback inhibition on the p53 pathway (*Manfredi, 2010*). The basal levels of Trp53inp1 and Mdm2 were also strongly reduced in TFEB/TFE3 DKO MEFs under etoposide treated conditions, indicating a novel role for TFE3 and TFEB in promoting their transcription in response to DNA damage.

Similarly, the pro-apoptotic p53 transcribed genes Bbc3 and Bax are significantly downregulated in TFEB/TFE3 DKO MEFs under etoposide treated conditions. Dram1 is another p53 induced transcript involved in regulation of both autophagy and apoptosis (*Crighton et al., 2006*). DRAM1 appears to promote apoptosis in a number of physiological contexts at least in part by promoting lysosomal membrane permeabilization (LMP) (*Guan et al., 2015*; *Laforge et al., 2013*). We found that both basal levels and etoposide induced levels of Dram1 mRNA are strongly reduced in TFEB/ TFE3 DKO MEFs.

Sesn1 and Sesn2 are highly homologous stress responsive genes that inhibit mTORC1 activity (*Cam et al., 2014*). They are also both direct transcriptional targets of p53, thus providing a direct mechanistic link between p53 activation in response to DNA damage and TFE3 and TFEB activation (*Budanov and Karin, 2008*). Sesn1 and Sesn2 were among the strongest downregulated targets in TFEB/TFE3 DKO MEFs in our RNA-Seq experiments. These data were recapitulated by direct qRT-PCR measurements with both genes exhibiting strong reductions in etoposide induced transcript levels, but only Sesn1 exhibiting a significant reduction under basal conditions.

Many other previously validated direct transcriptional p53 target genes, including Tp53 itself, were identified in the RNA-Seq data set and subsequently confirmed to be down-regulated in TFE3/ TFEB DKO MEFs in response to DNA damage (*Riley et al., 2008*). These include Cdkn1a, an extensively characterized cell-cycle control gene; Laptm5, a gene involved in LMP; the lysosomal hydrolase encoding gene Ctsd and Wrap53, an antisense Tp53 transcript required for Tp53 induction, among others (*Figure 3*) (*Inoue et al., 2009*; *Mahmoudi et al., 2009*; *Pommier et al., 2010*; *Renault et al., 2011*).

In order to test if these TFE3 and TFEB dependent defects in DDR and p53 related gene transcription are a conserved in other cell types, we decided to test an independently generated TFEB/ TFE3 DKO RAW264.7 macrophage cell line used in a previous study (*Pastore et al., 2016*). Indeed, the etoposide induced expression of numerous DDR response genes, including Bbc3, Trp53inp1,

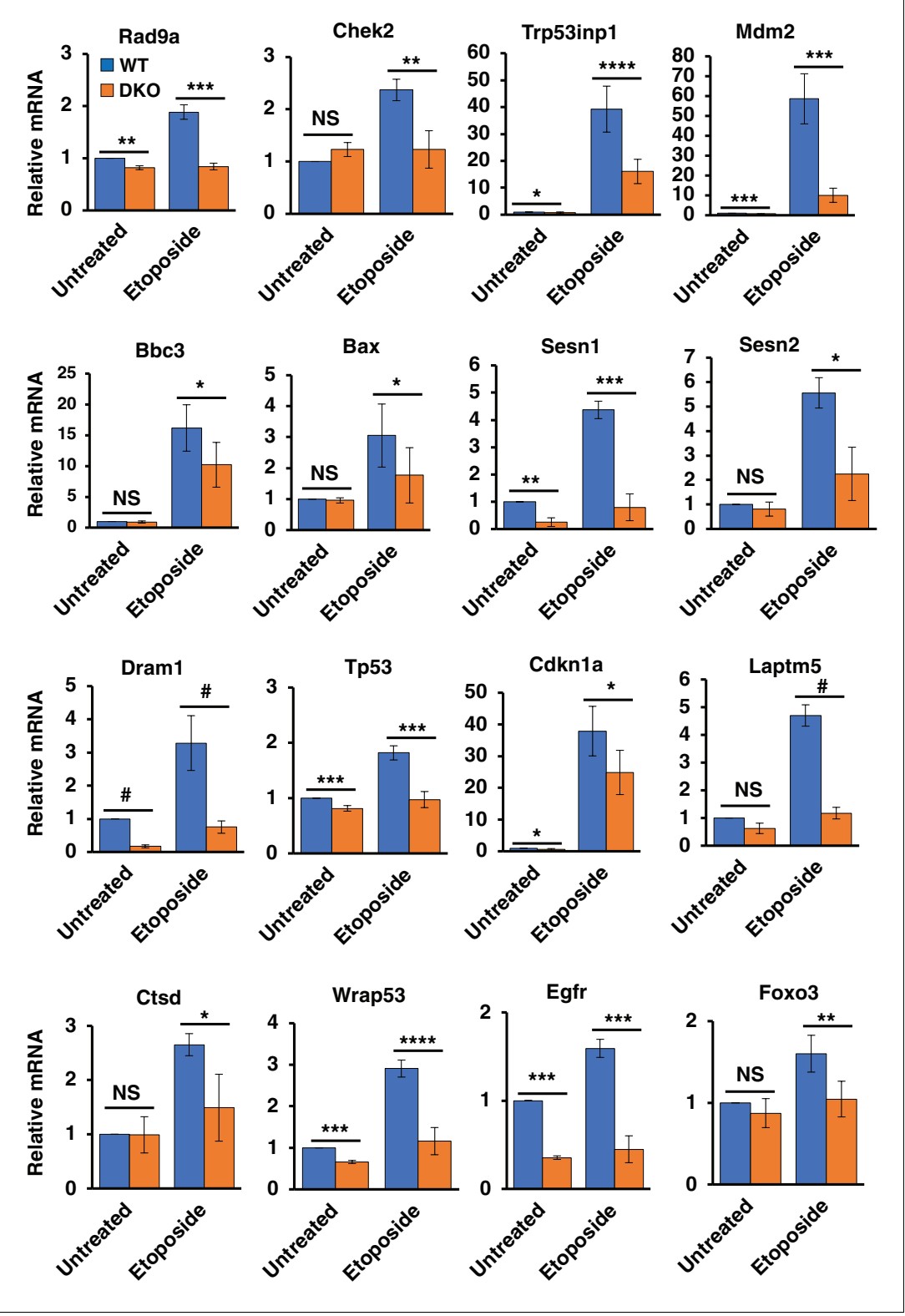

**Figure 3.** Differentially regulated genes in WT versus TFEB/TFE3 DKO MEFs undergoing DNA damage. qRT-PCR-based quantification of basal and etoposide induced mRNA levels of Rad9a, Chek2, Trp53inp1, Mdm2, Bbc3, Bax, Sesn1, Sesn2, Dram1, Tp53, Cdkn1a, Laptm5, Ctsd, Wrap53, Egfr and Foxo3 in WT vs TFEB/TFE3 DKO MEFs. All qRT-PCR data represented as geometric mean ± standard deviation and significance tested using Student's t-test (*p < 0.05, **p < 0.01, ***p < 0.001, ****p < 0.0001, #p < 0.00001).

*Figure 3 continued on next page*

*Figure 3 continued*

DOI: https://doi.org/10.7554/eLife.40856.012
The following source data and figure supplements are available for figure 3:
**Source data 1.** Differentially regulated genes in WT and TFEB/TFE3 DKOs.
DOI: https://doi.org/10.7554/eLife.40856.013
**Figure supplement 1.** qRT-PCR analysis of differentially expressed p53 upstream regulators and downstream effectors in WT versus TFEB/TFE3 DKO RAW264.7 cells.
DOI: https://doi.org/10.7554/eLife.40856.014
**Figure supplement 1—source data 1.** qPCR analysis of differentially expressed p53 upstrem regulators and downstream effectors.
DOI: https://doi.org/10.7554/eLife.40856.015

Sesn1, Sesn2, Chek2, Rad9a, Foxo3, Mdm2, and Wrap53 were strongly downregulated in TFEB/ TFE3 DKO RAW264.7 macrophages (*Figure 3—figure supplement 1*; numerical values *Figure 3-figure supplement 1-source data 1* and *Supplementary file 2*).

## TFE3 and TFEB help sustain p53 protein levels in response to DNA damage

One possible explanation for the reduced p53-dependent transcription observed in TFEB/TFE3 DKO cells is that TFE3 and TFEB play a role in p53 signaling in response to DNA damage. To test this hypothesis, we compared p53 activation and stabilization in WT vs TFEB/TFE3 DKO RAW264.7 cells. These cells were chosen because they exhibit a clear and highly reproducible p53 induction pattern in response to DNA damage. Additionally, SV40 immortalization of the MEFs used in this study can affect p53 stability, causing confounding effects. As expected, WT RAW264.7 exhibited a rapid and sustained increase in total p53 levels and levels of phosphorylation at serine 15. This was correlated with a concomitant increase in the protein levels of the p53 E3 ubiquitin ligase, Mdm2 starting at 2 hr etoposide treatment (*Figure 4A and B*, see also Figure 9E). Like the WT cells, TFEB/TFE3 DKO RAW264.7 showed a rapid induction of total and phospho-serine 15 p53 until 2 hr etoposide treatment. At this point, total and phospho-serine 15 p53 levels declined at a steady state until they reached close to baseline levels by 8 hr etoposide treatment (*Figure 4A and B*; numerical values *Figure 4—source data 1*). Furthermore, the levels of Mdm2 were significantly higher in the TFEB/TFE3 DKO cells at 2 and 4 hr etoposide treatment, precisely when the levels of p53 begin to diverge between the two genotypes (*Figure 4A and C*; numerical values *Figure 4—source data 2*). These observations indicate that TFE3 and TFEB are dispensable for the initial activation and stabilization of p53, however their presence is required for a robust and sustained p53 response after prolonged DNA damage, possibly through suppression of Mdm2 protein levels. Given these data, we hypothesized that the presence of TFE3 and TFEB may extend the half-life of p53 under DNA damage conditions. To test this, p53 was induced in both WT and TFEB/TFE3 DKO RAW264.7 cells by treating cells for 2 hr with etoposide, followed by an 8 hr cycloheximide chase, also in the presence of etoposide. As seen in *Figure 4D and E*; numerical values *Figure 4—source data 3*, p53 protein levels were elevated for a longer time in WT compared to DKO cells, consistent with our hypothesis. In order to test if the decrease in p53 half-life and protein levels observed in DKO cells after extended periods of DNA damage was due to elevated Mdm2 levels, we assessed p53 protein levels in both cell types after 8 hr treatment with etoposide in the presence and absence of the Mdm2 specific inhibitor, nutlin-3. Nutlin-3 treatment modestly increased the levels of p53 observed in WT RAW264.7 cells. In contrast, nutlin-3 drastically increased the levels of p53 in RAW264.7 TFEB/TFE3 DKO cells, bringing its levels on par with those observed in WT cells (*Figure 4F and G*; numerical values *Figure 4—source data 4*). No apparent differences in the ratio of phospho-Ser166 modified Mdm2 to total Mdm2 were detected under any conditions, suggesting that activation of Mdm2 is unaffected by TFEB/TFE3 status (*Figure 4F*).

## Over-expression of TFEB and TFE3 increases p53 protein stability

Since loss of TFEB and TFE3 was found to affect p53 transcriptional activity and stability in various TFEB/TFE3 knockout cell lines, we predicted that overexpression of constitutively active mutants of

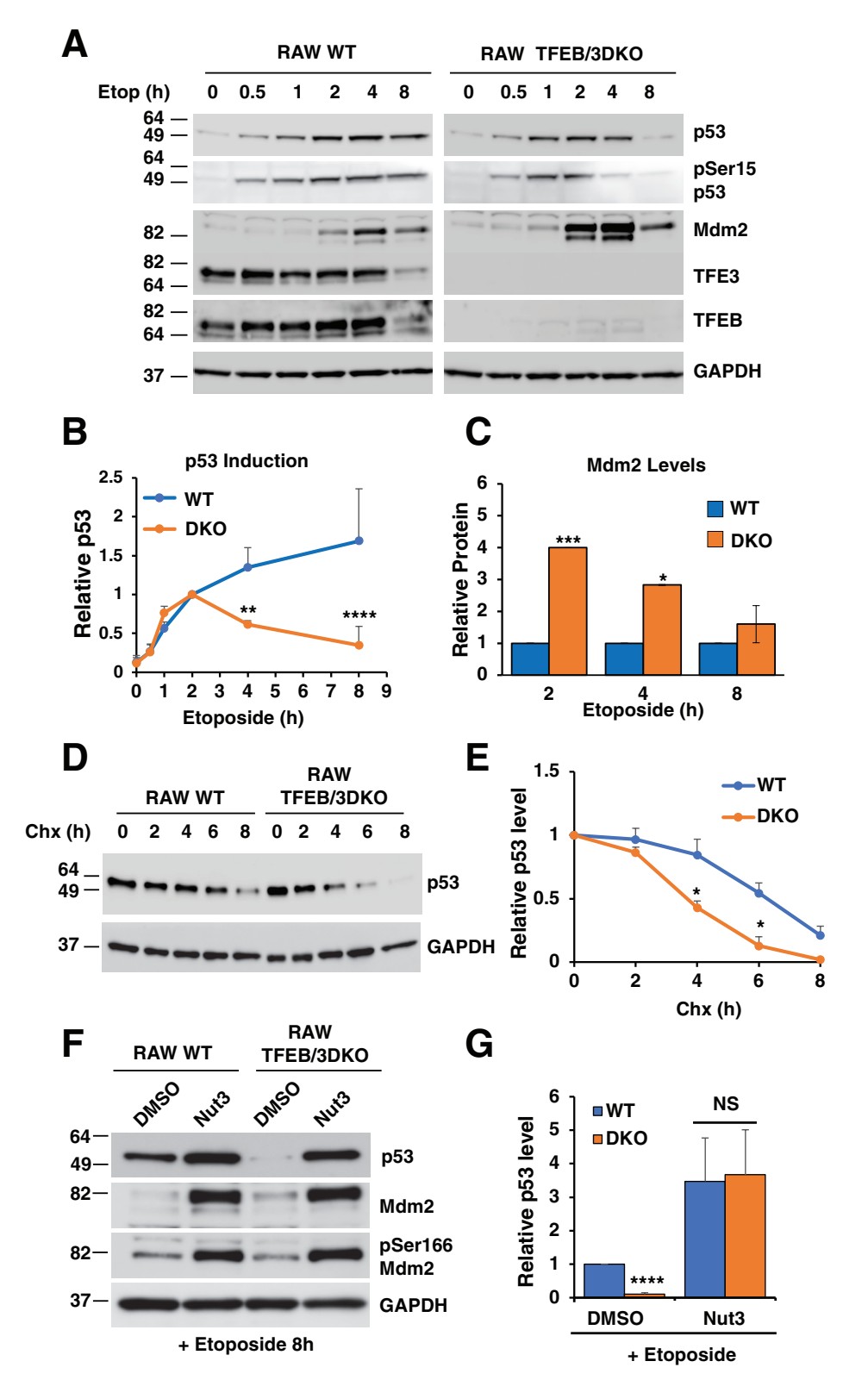

**Figure 4.** p53 induction in response to DNA damage is impaired in TFEB/TFE3 DKO RAW264.7. 7 cells. (**A**) Representative Western blot showing p53 induction, p53 Ser15 phosphorylation, and Mdm2 levels in WT and TFE3/TFEB DKO RAW264.7 cells following etoposide treatment up to 8 hr. (**B**) Quantification of p53 induction from data shown in A. Data represents mean relative p53 level ± standard deviation with n = 3. Significance tested with two-way ANOVA with Sidak's multiple comparisons test (**p < 0.01, ****p < 0.0001). (**C**) Quantification of data shown in A. Total Mdm2 levels are

*Figure 4 continued on next page*

*Figure 4 continued*

significantly increased in TFEB/TFE3 DKO RAW264.7 cells compared to WT controls at 2- and 4 hr etoposide treatment. Data represents mean relative Mdm2 levels ± standard deviation with n=3. Significance tested with Student's test (*p < 0.05, ***p < 0.001). (**D**) Representative Western blot of cycloheximide chase assay showing decreased p53 half-life in TFEB/TFE3 DKO RAW264.7 cells compared to WT controls. Cells were pre-treated with etoposide 2 hr to induce p53 expression and were chased in the presence of etoposide and cycloheximide. (**E**) Quantification of p53 levels from data shown in D. Data represents mean relative p53 level ± standard deviation with n = 3. Significance tested with Student's test (*p < 0.05). (**F**) Representative Western blot showing rescue of p53 expression levels by treatment with nutlin-3 in TFEB/TFE3 DKO RAW264.7 cells after 8 hr etoposide treatment. (**G**) Quantification of p53 levels shown in F. Data represents mean relative p53 level ± standard deviation with n = 3. Significance tested using Student's t-test (****p < 0.0001).
DOI: https://doi.org/10.7554/eLife.40856.016

The following source data is available for figure 4:

**Source data 1.** Quantification of p53 induction in WT and TFEB/TFE3 DKO Raw264.7 cells.
DOI: https://doi.org/10.7554/eLife.40856.017
**Source data 2.** Quantification of Mdm2 levles in WT and TFEB/TFE3 DKO Raw264.7 cells.
DOI: https://doi.org/10.7554/eLife.40856.018
**Source data 3.** Quantification of p53 levels in chx-treated WT and TFEB/TFE3 DKO Raw264.7 cells.
DOI: https://doi.org/10.7554/eLife.40856.019
**Source data 4.** Quantification of p53 levels in Nut3-treated WT and TFEB/TFE3 DKO Raw264.7 cells.
DOI: https://doi.org/10.7554/eLife.40856.020

TFEB and TFE3 should be sufficient to increase basal p53 expression levels in certain cellular contexts. HeLa cells were chosen to test this hypothesis due to their extremely low levels of basal p53 expression and their ability to robustly over-express high levels of exogenous protein through adenovirus transduction. Overexpression of the constitutively active forms of TFEB (TFEB-S211A) and TFE3 (TFE3-S321A) dramatically increased the basal levels of p53 observed in whole cell lysates (*Figure 5A and B*; numerical values *Figure 5—source data 1*). This elevation in p53 protein level was further increased in the presence of etoposide, suggesting a synergistic effect between activated TFE3 or TFEB in response to DNA damage (*Figure 5A and B*). Under our experimental conditions, endogenous p53 is barely detectable under basal conditions in HeLa cells when assessed via immunofluorescence. However, expression of TFEB-S211A and TFE3-S321A led to a near uniform pattern of p53 expression, predominantly localized in the nuclei of HeLa cells (*Figure 5C*). Given our data in RAW264.7 cells suggesting that loss of TFE3 and TFEB decreases p53 half-life, we performed a complimentary cycloheximide chase experiment to test if expression of TFEB-S211A and TFE3-S321A could increase the half-life of p53. The half-life of endogenous p53 is less than 15 min in HeLa cells under basal conditions. This was extended beyond 60 min in TFEB-S211A expressing cells and 30 min in TFE3-S321A expressing cells (*Figure 5D and E*; numerical values *Figure 5—source data 2*). Quantification of these experiments showed that TFEB-S211A expression significantly increased the relative p53 levels at all cycloheximide chase time points from 15 min to 90 min, while TFE3-S321A expression significantly increased the relative p53 after 30 min cycloheximide chase (*Figure 5E*). The relatively stronger effect of TFEB-S211A on p53 levels compared to TFE3-S321 may be due to a somewhat higher nuclear localization and stability of the TFEB construct compared to the TFE3 construct.

In order to rule out that the adenovirus-mediated expression of TFEB-S211A and TFE3-S321A and subsequent increase in p53 protein levels was not due to some non-specific viral effect or due to excessive exogenous protein load, we expressed flag tagged TFEB-S211A from a plasmid which results in significantly lower total expression and a more heterogeneous cellular distribution after transfection. We also expressed a nuclear localization mutated construct, TFEB-S211A/NLSmut. Transfection of HeLa cells with TFEB-S211A, but not TFEB-S211A/NLSmut increased basal p53 protein levels, despite similar expression levels of both constructs (*Figure 5—figure supplement 1A*). This indicates that TFEB nuclear localization is required for p53 stabilization and this effect is therefore likely due to TFEB transcriptional activity. Immunofluorescence of HeLa cells expressing plasmid encoded TFEB-S211A shows a clear correlation between transfected cells and p53 nuclear accumulation, with untransfected cells exhibiting undetectable p53 (*Figure 5—figure supplement 1B*). Consistent with our analysis of p53 protein levels in cell lysates, expression of TFEB-S211A/NLSmut

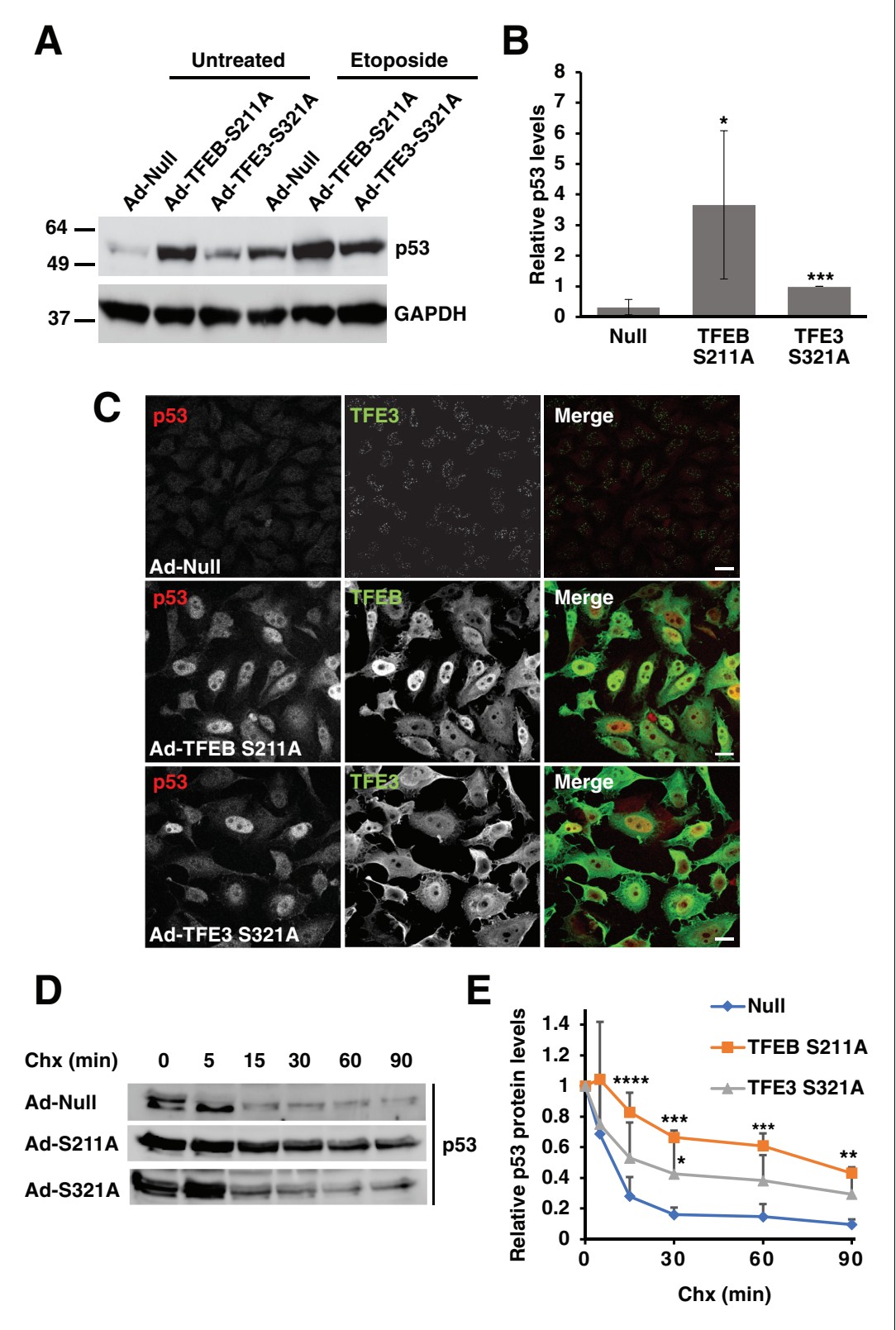

**Figure 5.** Expression of constitutively active TFEB and TFE3 in HeLa cells increases total p53 protein levels and its extends half-life. (**A**) Representative Western blot showing elevated p53 protein levels in adenovirus infected HeLa cells expressing constitutively active mutants of TFEB and TFE3. Further p53 protein level increases are seen
*Figure 5 continued on next page*

*Figure 5 continued*

with constitutively active TFEB and TFE3 after treatment with etoposide 8 hr. (**B**) Quantification of basal p53 protein level in HeLa cells expressing constitutively active TFEB and TFE3. Due to the high dynamic range and low detectability of basal endogenous p53 in control cells, values were normalized to intermediate expression samples, TFE3 S321A. Data represents mean relative p53 level ± standard deviation with n = 5 (*p < 0.05, ***p < ). (**C**) Immunofluorescence images of HeLa cells expressing constitutively active TFEB and TFE3 exhibit robust p53 accumulation in the nucleus compared to control cells. Scale bar = 20 μm. (**D**) Representative Western blot of cycloheximide chase assay showing increased stability of p53 in HeLa cells infected with control (Null) adenovirus or adenovirus expressing constitutively active TFEB and TFE3. (**E**) Quantification of cycloheximide chase assay shown in D. Values represent mean p53 protein levels ± standard deviation and each condition normalized relative to time 0 for that cell population with n = 4. Significance tested with two-way ANOVA with Dunnett's multiple comparisons test (*p < 0.05, **p < 0.01, ***p < 0.001, ****p < 0.0001).
DOI: https://doi.org/10.7554/eLife.40856.021

The following source data and figure supplements are available for figure 5:

**Source data 1.** Quantification of p53 levels in cells transfected with TFEB and TFE3 active mutants.
DOI: https://doi.org/10.7554/eLife.40856.022
**Source data 2.** Quantification of p53 levels in chx-treated HeLa cells.
DOI: https://doi.org/10.7554/eLife.40856.023
**Figure supplement 1.** (A) Western blot showing that TFEB-S211A expression, but not the NLS mutant, increases p53 protein levels in HeLa cells.
DOI: https://doi.org/10.7554/eLife.40856.024
**Figure supplement 1—source data 1.** qPRC analysis of DDR and p53-dependent gene expression.
DOI: https://doi.org/10.7554/eLife.40856.025

failed to induce a strong accumulation of p53 when assessed by immunofluorescence (*Figure 5—figure supplement 1B*).

Since loss of TFE3 and TFEB leads to defects in DDR genes and p53 related signaling in various cell types, we expect to see an increase in expression of some of these genes in response to overexpression of constitutively active TFE3 and TFEB. Adenovirus mediated expression of TFEB-S211A and TFE3-S321A in HeLa cells lead to a significantly increased expression of the basal levels of a number of these genes, including Bbc3, Trp53inp1, Sesn1, UVRAG and Rad9A (*Figure 5—figure supplement 1C*; numerical values *Figure 5—figure supplement 1—source data 1*). Additionally, the p53 regulatory genes Ube2b and Ube4b, which encode p53 E2 and E4 ubiquitin ligases were also strongly upregulated by TFEB-S211A and S321A. (*Figure 5—figure supplement 1C*). Importantly, over-expression of TFEB-S211A and S321A did not cause Tp53 transcriptional upregulation, further suggesting a role of TFEB and TFE3 in promoting p53 protein stabilization (*Figure 5—figure supplement 1C*).

## DNA damage induces lysosomal membrane permeabilization in a TFE3 and TFEB dependent manner

LMP has long been known to be a downstream effect of p53 activation (*Yuan et al., 2002*). LMP occurs when the integrity of the lysosomal limiting membrane is disrupted, allowing the release of lysosomal hydrolases into the cytosol and may result in either apoptosis or necrosis, depending on the nature and extent of the lysosomal permeabilizing stimulus and the cell type involved (*Repnik et al., 2014*).

Given that TFE3 and TFEB have been previously linked to apoptotic regulation (*Martina et al., 2016*) and that expression of Dram1 and Laptm5, two p53-induced genes implicated in LMP and apoptosis, is strongly downregulated in TFEB/TFE3 cells, we hypothesized that TFE3 and TFEB may exert some of their apoptotic activity through LMP. To measure LMP, we used an immunofluorescence galectin puncta assay which measures galectin-1 localization with LAMP1. Under normal conditions, galectin-1, a β-galactoside binding protein, is localized throughout the cytosol and nucleus. After LMP, galectin-1 can pass through the lysosomal limiting membrane and bind to the carbohydrate rich glycocalyx within the lysosomal lumen. In WT MEFs, etoposide treatment resulted in a time dependent increase in LMP, with most cells exhibiting significant LMP by 8 hr, while untreated MEFs exhibit extremely low levels of detectable LMP (*Figure 6A–D*; numerical values *Figure 6—*

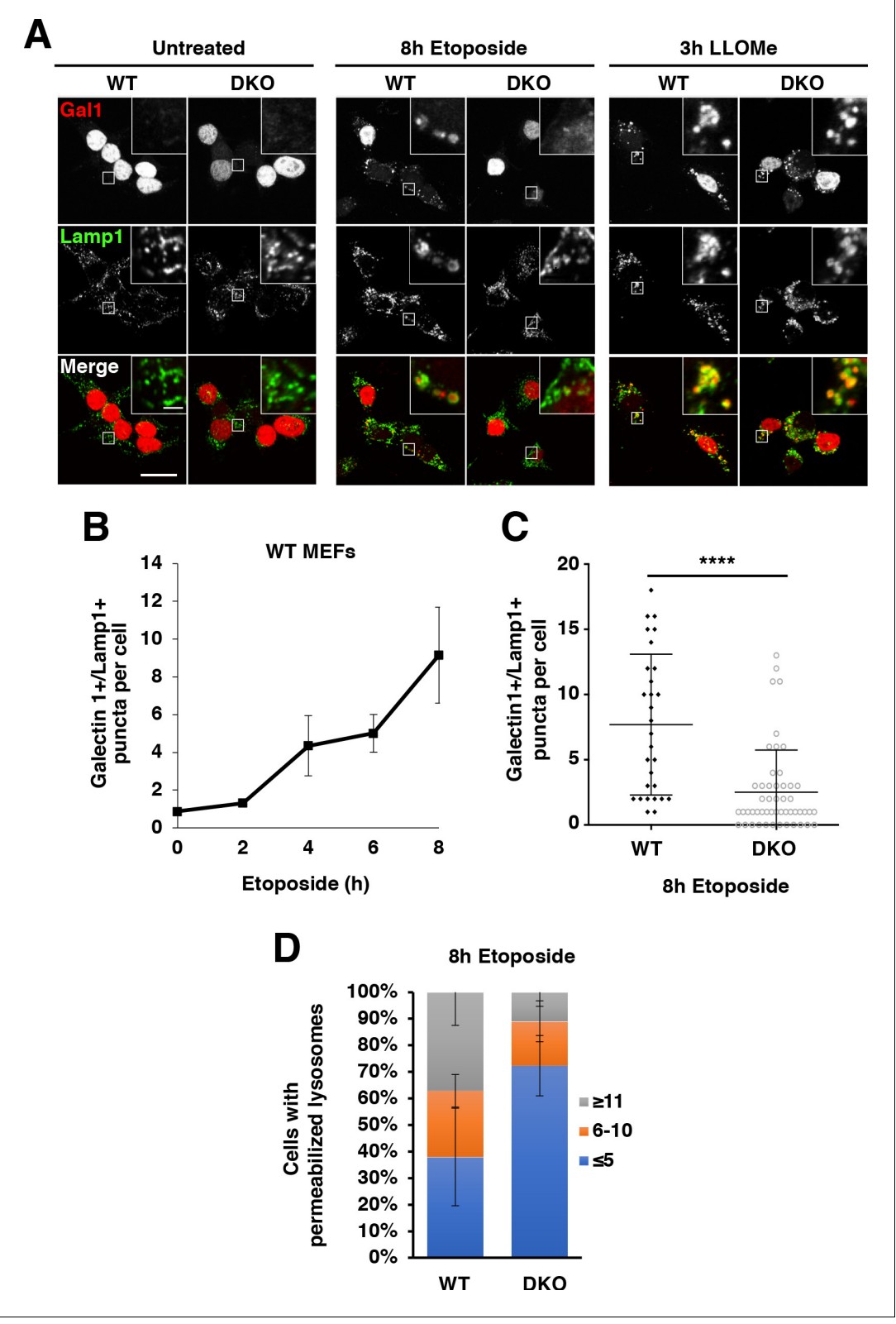

**Figure 6.** TFEB and TFE3 are essential for etoposide-induced lysosomal membrane permeabilization in MEFs. (A) Immunofluorescence images showing LMP in MEFs. Red galectin-1 puncta appear co-localized or within the lumen of green Lamp1 positive lysosomes. No LMP is detected under basal conditions in either WT or TFEB/TFE3 DKO MEFs. Treatment with etoposide induces profound LMP in WT, but not TFEB/TFE3 DKO cells. No differences in

*Figure 6 continued on next page*

*Figure 6 continued*

LMP induction were detected in LLOMe treated cells, regardless of genotype. Scale bar = 20 µm, inset = 2 µm. (**B**) WT MEFs exhibit a time-dependent increase in LMP after etoposide treatment. Quantification of data shown in A of galectin-1+/Lamp1 + LMP puncta per WT MEF cell. Data represent mean number of puncta per cell ± standard deviation from randomly selected confocal images, with > 20 cells per counted for each time point over three separate experiments. (**C**) Quantification of total number of galectin-1+/Lamp1+ LMP puncta per cell in WT vs TFE3/TFEB DKO MEFs treated for 8 hr with etoposide. Distribution is representative of one of the three independent experiments performed and shows 29 randomly selected WT MEF cells and 51 randomly selected TFEB/TFE3 DKO MEF cells. Significance determined using Student's t-test (****p < 0.0001). (**D**) Quantification of overall galectin-1+/Lamp1+ LMP puncta distribution after 8 hr etoposide treatment in WT versus TFE3/TFEB DKO MEFs. Data were binned from three separate experiments with > 100 cells represented in each category and a minimum of 29 cells from each trial.

DOI: https://doi.org/10.7554/eLife.40856.026

The following source data is available for figure 6:

**Source data 1.** Quantification of LMP following etoposide treatment.
DOI: https://doi.org/10.7554/eLife.40856.027

**Source data 2.** Quantification of galectin-1/lamp1-positive puncta in WT and DKO MEFs treated with etoposide.
DOI: https://doi.org/10.7554/eLife.40856.028

---

*source datas 1* and *2*). In TFEB/TFE3 DKO MEFs, a significantly reduced proportion of cells exhibit LMP by 8 hr etoposide, and those that do, had fewer galectin-1 positive puncta on average than control cells (*Figure 6A* and *Figure 6C,D*). The small molecule, LLOMe, accumulates within lysosomes where it exhibits direct membranolytic activity (*Repnik et al., 2014*). In contrast to etoposide treatment, LLOMe treated MEFs exhibited extensive and equivalent degrees of LMP activity in both WT and DKO backgrounds (*Figure 6A*). This indicates that DKO MEFs are physically capable of inducing LMP, however TFE3 and TFEB participate in induction of LMP in response to DNA damage.

## Apoptotic cell death due to DNA damage is facilitated by TFE3 and TFEB

We have previously shown that TFEB/TFE3 DKO MEFs exhibit delayed cell death in response to ER-Stress (*Martina et al., 2016*). In order to test if TFE3 and TFEB contribute to canonically p53-dependent, DNA damage induced apoptosis, we tested the viability of RAW264.7 cells exposed to etoposide.

First, we looked at Caspase-3 cleavage, a biochemical indicator of apoptosis. Both WT and TFEB/TFE3 DKO RAW264.7 cells exhibited detectable cleaved Caspase-3 after 8 hr etoposide treatment (*Figure 7A*). However, WT RAW264.7 cells exhibited a sustained and much higher degree of detectable Caspase-3 after prolonged etoposide treatment for 16 and 24 hr, which was highly significant at 16 hr (*Figure 7A and B*; numerical values *Figure 7—source data 1*), suggesting that TFE3 and TFEB are critical for the proper induction of DNA damage induced apoptosis. Interestingly, LMP has been shown to occur downstream of caspase activation in etoposide treated cells and LMP in turn has been shown to promote the apoptotic cell death, indicating that the LMP defects and apoptotic defects observed in our studies may be linked as well (*Oberle et al., 2010*).

Early in apoptosis, cells exhibit increased phosphatidylserine on the outer leaflet of the plasma membrane. This can be quantified with flow cytometry by measuring the level of Annexin V bound to the cell surface. Cell death can similarly be quantified by assessing the integrity of the plasma membrane through detection of the membrane impermeant dye, 7-AAD within the cell population. For this assay, cells negative for both Annexin V and 7-AAD (Annexin V-/7-AAD-) are considered live, cells positive for Annexin V and negative for 7-AAD (Annexin V+/7-AAD-) are indicative of early apoptosis, cells positive for both Annexin V and 7-AAD (Annexin V+/7-AAD+) are undergoing late apoptosis, and cells negative for Annexin V and positive for 7-AAD (Annexin V-/7-AAD+) are exhibiting necrosis-like characteristics. WT and TFEB/TFE3 DKO RAW264.7 cells both exhibit high viability under un-stressed conditions with minimal apoptosis detected (*Figure 7C and D*; numerical values *Figure 7—source data 2*). After 16 and 24 hr etoposide treatment, major differences in cell viability were observed with significantly more live TFEB/TFE3 DKO compared to RAW264.7 WT cells as evidenced by the number of AnnexinV-/7-AAD- cells detected. This is paralleled with a significant

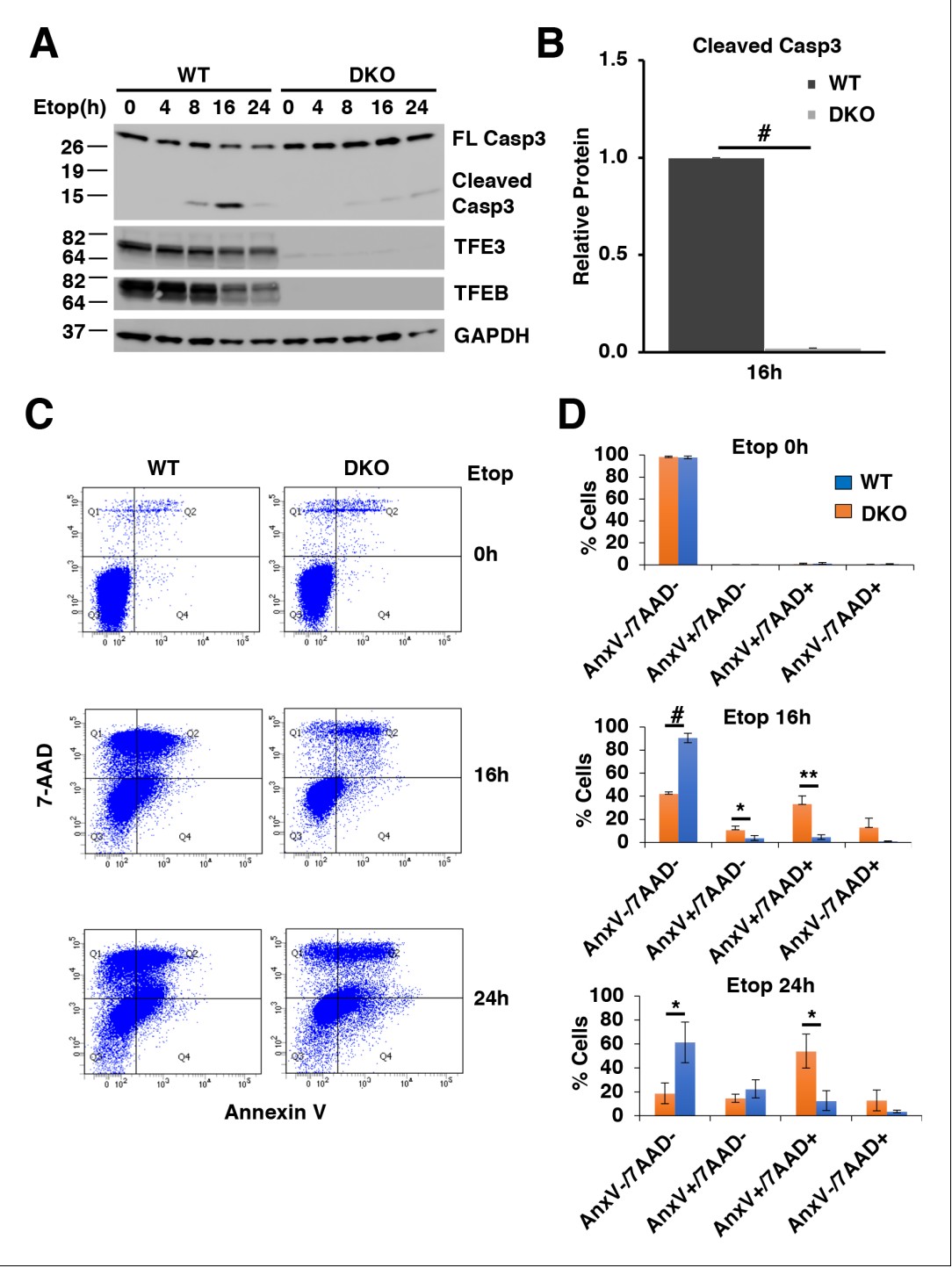

**Figure 7.** TFE3 and TFEB are necessary for proper execution of apoptosis in response to DNA damage in RAW264.7 cells. (A) Representative Western blot showing Caspase-3 cleavage in response to increasing time of etoposide treatment. (B) Quantification of data shown in A indicating defects in Caspase-3 cleavage in TFEB/TFE3 DKO RAW264.7 cells. Cleaved Caspase-3 levels are normalized to WT cells after 16 hr etoposide treatment with n = 3. Significance tested with Student's t-test (#p < 0.00001). (C) Annexin V/7-AAD flow cytometry assay data showing an impaired progression through early (AnnexinV+/7-AAD-) and late (AnnexinV+/7-AAD+) apoptosis in TFEB/TFE3 DKO RAW264.7 cells after 16 and 24 hr etoposide treatment. (D) Quantification of data shown in C. Data taken from three independent experiments and significance tested using Student's t-test (*p < 0.05, **p < 0.01, #p < 0.00001).

*Figure 7 continued on next page*

*Figure 7 continued*

DOI: https://doi.org/10.7554/eLife.40856.029

The following source data is available for figure 7:

**Source data 1.** Quantification of cleaved Caspase-3 levels.

DOI: https://doi.org/10.7554/eLife.40856.030

**Source data 2.** Quantification of AnnexinV/7AAD levels by flow cytometry assays.

DOI: https://doi.org/10.7554/eLife.40856.031

increase in the number of Annexin $V^+$/7-AAD$^-$ (early apoptotic) and/or Annexin $V^+$/7-AAD$^+$ (late apoptotic) in WT RAW264.7 cells compared to TFEB/TFE3 DKO cells at 16 hr and 24 hr etoposide treatment, respectively (*Figure 7C and D*). These results indicate that TFE3 and TFEB are indispensable for efficient apoptosis induction in response to DNA damage.

## Altered cell cycle regulation in TFEB/TFE3 knockout cells

The comparative transcriptome analysis between etoposide-treated WT and TFEB/TFE3-DKO RAW264.7 cells revealed that close to 5000 genes were differentially express between both cell lines (*Figure 8A and B*, *Supplementary file 2*). Gene ontology (GO) analysis of 'Biological Process' terms showed a very strong down-regulation of genes implicated in response to DNA damage and DNA repair (*Figure 8—figure supplement 1A*). This was to be expected given the reduced p53 levels in DKO cells. More surprising was the significant downregulation of genes implicated in cell cycle regulation (*Figure 8—figure supplement 1A*). In fact, the most significantly downregulated categories were those related to mitosis, including 'M phase', 'M phase of mitotic cell cycle', 'nuclear division', 'mitosis', 'organelle fission', 'chromosome segregation', 'mitotic cell cycle' and 'cell division'. Furthermore, the heatmap of the 50 most significantly dysregulated genes in etoposide-treated TFEB/TFE3 DKOs included several critical regulators of the cell cycle, including Ccnb1 (Cyclin B1), Ccnb2 (Cyclin B2), Ccna2 (Cyclin A2), AurkB (Aurora kinase B), Birc5 (Survivin), Plk1 (Polo like kinase 1) and Ttk (Ttk protein kinase), as well as genes implicated in chromosome segregation and cytokinesis such as Nuf2 (NDC80 kinetochore complex component), Prc1 (protein regulator of cytokinesis 1), Nusap1 (nucleolar and spindle associated protein 1), Cenpe (centromere protein E), Esco2 (establishment of sister chromatid cohesion N-acetyltransferase 2), Cep55 (centrosomal protein of 55 kDa), Espl1 (extra spindles poles-like 1), Knl1 (kinetochore scaffold 1) and Spag5 (astrin) (*Figure 8—figure supplement 1B*).

Recent evidence suggests that the inability to induce apoptosis in conditions of persistent DNA damage may lead to senescence as an irreversible cell cycle arrest (*Qian and Chen, 2010*). The primary mechanism for G2/M cell cycle arrest triggered by p53 is the p21-mediated stabilization of the dimerization partner, RB-like, E2F and multi-vulval class B (DREAM) complex, a transcriptional repressor complex which is central to the downregulation of cell cycle-related genes (*Quaas et al., 2012*). In parallel, p21 also inhibits CDK-dependent phosphorylation of the retinoblastoma protein (Rb) to keep E2F-regulated genes in an inactive state, thus preventing the transition from the G1 phase to the DNA synthesis S phase (*Niculescu et al., 1998*; *Xiong et al., 1993*).

The DREAM pathway coordinately downregulates more than 250 genes implicated in DNA replication, mitotic spindle assembly, nucleosome packaging and chromosome segregation, among others (*Engeland, 2018*). Therefore, we decided to compare expression of DREAM targets between WT and TFEB/TFE3 DKO cells. As seen in *Figure 8*, no major differences were observed in the expression of cell cycle regulators between WT and DKOs under control conditions (*Figure 8C*, *Supplementary file 3*). In contrast, 91% (240 out of 262) of the genes regulated by the DREAM pathway showed a much more significant downregulation in TFEB/TFE3 DKO than in WT cells following etoposide treatment (*Figure 8C*, *Supplementary file 3*). These results indicate that the DREAM complex remains hyperactive in TFEB/TFE3-depleted cells in response to genotoxic stress.

The hyperactivation of the DREAM complex may be a consequence of the inability of the TFEB/TFE3 DKO cells to induce apoptosis in response to prolonged DNA damage. Alternatively, TFEB and TFE3 might have a more direct role in the regulation of cell cycle checkpoints in response to stress. We have previously performed ChIP-seq analysis for TFE3 in RAW 264.7 cells subjected to different stress conditions (*Pastore et al., 2016*). To investigate whether TFEB and TFE3 may have a

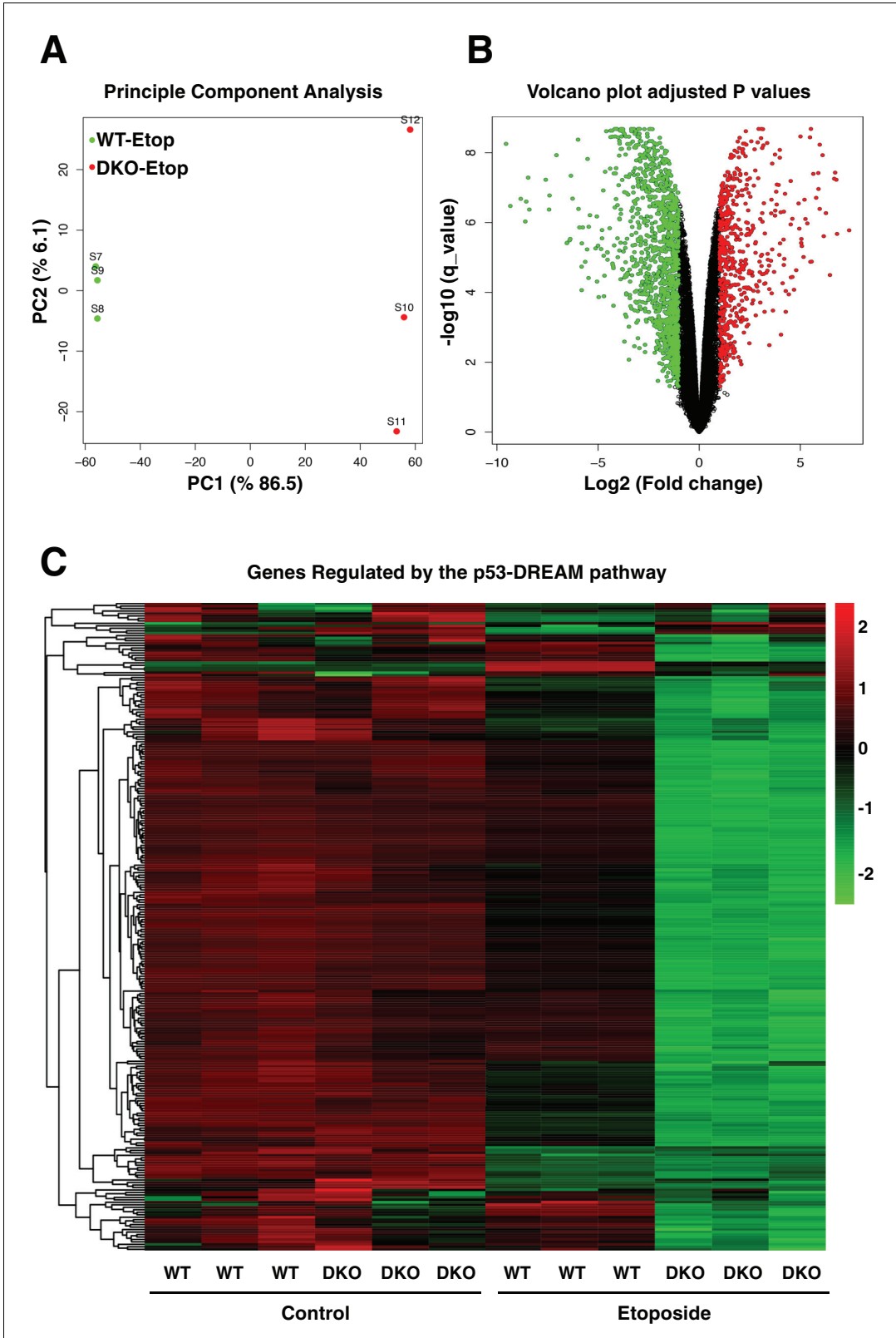

**Figure 8.** Comparative gene expression of etoposide-treated WT and TFEB/TFE3-DKO RAW264.7 cells. (**A**) Principal component analysis of genes with q-value < 0.05 reveals distinct clustering of WT and TFEB/TFE3 DKO RAW 264.7 cells exposed to etoposide. (**B**) Volcano plot indicating distribution of genes significantly down- and up-regulated in WT versus TFEB/TFE3 DKO RAW264.7 cells exposed to etoposide for 8 hr. Cutoffs indicate genes with q-value < 0.05. (**C**) Hierarchical cluster heat map showing expression of DREAM pathway genes in three independent samples of WT and TFEB/TFE3

*Figure 8 continued on next page*

*Figure 8 continued*

DKO RAW264.7 cells following etoposide treatment for 8 hr. Each row shows the relative expression level of a single mRNA. Each column shows the expression level of a single sample. Up-regulated mRNAs are shown in red and down-regulated mRNAs are shown in green.

DOI: https://doi.org/10.7554/eLife.40856.032

The following figure supplement is available for figure 8:

**Figure supplement 1.** (A) Enriched GO terms in the 'Biological Process' category of differentially expressed genes between etoposide-treated WT and TFEB/TFE3 DKO RAW264.7 cells.

DOI: https://doi.org/10.7554/eLife.40856.033

direct role in cell cycle regulation, we reanalyzed our data in search for key cell cycle regulators which expression might be directly regulated by these transcription factors. Interestingly, we observed increased TFE3 binding to the promoter of CDK4 and CDK7 following TFE3 activation (*Figure 9A*). CDK4 plays a critical role in Rb inactivation and disassembly of the DREAM complex, while CDK7 functions as a CDK-activating kinase (CAK) by directly phosphorylating several of the CDKs and directly controlling cell-cycle progression (*Guiley et al., 2015*; *Malumbres, 2014*). To confirm that TFEB/TFE3 regulate CDK4 and CDK7 expression, we expressed constitutive active versions of TFEB and TFE3 in HeLa cells. As seen in *Figure 9B–C*; numerical values *Figure 9—source datas 1* and *2*, over-expression of either TFEB or TFE3 resulted in a significant increase in CDK4 and CDK7 mRNA and protein levels. Furthermore, phosphorylation of Rb, a CDK4 target, was significantly reduced in TFEB/TEF3 DKO cells (*Figure 9D–F*; numerical values *Figure 9—source data 3*). We also observed reduced RBL2 phosphorylation in TFEB/TFE3-depleted cells (*Figure 9D*). It has been recently described that phosphorylation of RBL2 by CDK4 is critical to weaken the activity of the DREAM complex (*Guiley et al., 2015*), offering a possible explanation for the observed hyperactivation of the DREAM complex in TFEB/TFE3 DKOs. Therefore, these results suggest that TFEB and TFE3 may have an important and previously unrecognized role in the regulation of the cell cycle checkpoints.

In summary, our data show that the cellular response to DNA damage is profoundly altered in the absence of TFEB and TFE3. The reduced stability of p53, enhanced cell cycle arrest, and inability to induce apoptosis under conditions of persistent genotoxic stress may push cells towards senescence and have important consequences in cancer progression.

## Discussion

TFEB and TFE3 are master transcriptional regulators of autophagy and lysosome biogenesis in response to starvation. However, recent studies have revealed new roles for these transcription factors in the control of a number of diverse biological functions, including the inflammatory process, the unfolded protein response and metabolic regulation. These different functions are often cell type or tissue specific and constitute a response to a variety of physiological stressors. Additionally, there is ample evidence that TFEB and TFE3 directly upregulate the expression of other transcription factors and co-activators involved in stress responsive pathways, such as ATF4 in ER stress and Pgc1a in metabolic stress (*Martina et al., 2016*; *Settembre et al., 2013*). Similarly, the tumor suppressor p53 represents a master transcriptional regulator of the DNA damage response which promotes cell-cycle arrest, DNA repair, and ultimately apoptosis (*Bunz et al., 1998*). Much like TFEB and TFE3, p53 has also been implicated in a number of additional biological processes, including autophagy, angiogenesis, metabolism, and cell migration in response to different stress conditions (*Bieging and Attardi, 2012*; *Tasdemir et al., 2008*; *Zhang et al., 2017*).

In this study, we present a two-part model in which p53 and TFE3/TFEB regulate each other's functions during different phases of the DNA damage response (*Figure 10*). After DNA damage events, p53 is rapidly activated to promote the transcription of numerous downstream targets involved in DNA repair, cell cycle arrest and apoptosis. Some of these targets of p53 transcription, including Sesn1 and Sesn2, inhibit mTORC1 signaling. We propose that in response to DNA damage, p53 activates TFEB and TFE3 through its inhibitory effects on mTORC1 activity. This p53-dependent activation of TFEB and TFE3 further enhances p53 signaling through both feedback and feedforward mechanisms. TFEB and TFE3 activation results in increased p53 stabilization and protein levels, resulting in enhanced expression of a number of transcripts involved in the DNA damage

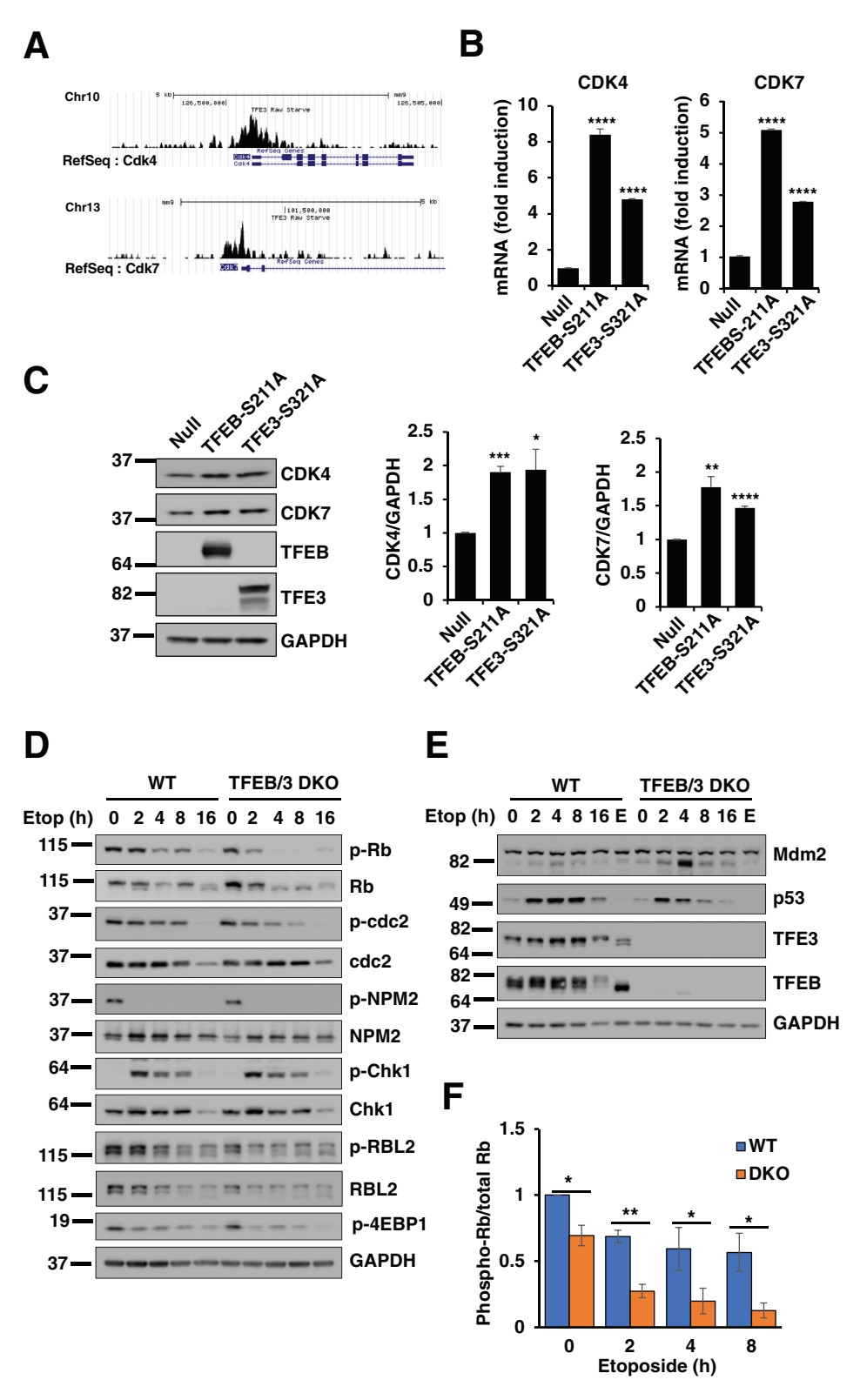

**Figure 9.** TFEB and TFE3 promote expression of cell cycle regulators. (A) Profiles of Chip-seq analysis for TFE3 in CDK4 and CDK7 promoters in RAW264.7 cells under stress condition. (B) qPCR-based quantification of CDK4 and CDK7 mRNA levels in adenovirus infected HeLa cells expressing control (Null) or constitutively active mutants of TFEB and TFE3. Data represented as geometric mean ± standard deviation and significance tested using Student's t-test with n = 3 (***p < 0.001, ****p < 0.0001). (C) Representative Western blot showing CDK4 and CDK7 levels in adenovirus infected

*Figure 9 continued on next page*

*Figure 9 continued*

HeLa cells expressing constitutively active mutants of TFEB and TFE3. Quantification of protein levels are shown on the right panels. Data represents mean relative CDK4 and CDK7 to GAPDH levels ± standard deviation. Significance tested using Student's t-test with n = 3 (*p < 0.05, **p < 0.01, ***p < 0.001, ****p < 0.0001). (D) Representative Western blot showing the expression of cell cycle regulators in response to etoposide treatment in WT versus TFEB/TFE3 DKO RAW264.7 cells. (E) Representative Western blot showing Mdm2 and p53 levels in WT and TFEB/TFE3 DKO RAW264.7 cells following etoposide treatment. EBSS, indicated as E, was used for 2 hr as a positive control for maximum mTORC1 inhibition. All the western blots are representative of three independent experiments. (F) Quantification of phospho-Rb/total-RB ratios from data shown in D. Data represents mean relative phospho-Rb to total Rb level ± standard deviation with n = 3. Significance tested using Student's t-test (*p < 0.05, **p < 0.01).

DOI: https://doi.org/10.7554/eLife.40856.034

The following source data is available for figure 9:

**Source data 1.** qPCR data showing CDK4 and CDK7 levels in cells expressing TFEB and TFE3 active mutants.

DOI: https://doi.org/10.7554/eLife.40856.035

**Source data 2.** Quantification of CDK4 and CDK7 protein levels.

DOI: https://doi.org/10.7554/eLife.40856.036

**Source data 3.** Quantification of phospho-RB/total-RB ratio.

DOI: https://doi.org/10.7554/eLife.40856.037

response. We hypothesize that when successful DNA repaired is achieved, activation of TFEB and TFE3 might contribute to cell survival, for example by promoting autophagy and re-initiation of the cell cycle. However, under conditions of prolonged DNA damage, like the ones presented in this study, TFEB and TFE3 ultimately facilitate apoptosis. TFEB/TFE3-mediated upregulation of Dram1, Laptm5 and other regulators of lysosome stability results in increased LMP. This LMP, combined with other pro-apoptotic targets of both p53 and TFEB/TFE3, such as Bbc3, and Bax, ultimately results in apoptotic cell death. In cells lacking TFEB and TFE3, the inability to induce efficient apoptosis in response to prolonged DNA damage results in the hyper-repression of multiple genes implicated in cell cycle and proliferation, a circumstance that may direct cells to senescence or quiescence and have important implications in cellular transformation.

The contribution of TFEB and TFE3 to cancer is probably complex and context dependent. It has long been documented that multiple translocation events, resulting in five described TFE3 gene-fusion products and one TFEB gene fusion product, result in renal cell carcinoma (RCC) (*Kauffman et al., 2014*). The manner in which these gene fusions promote oncogenesis is poorly understood, but it is thought that TFE3 and TFEB gene fusion products may upregulate oncogenic pathways already active in the cell such as the Wnt-beta-catenin and mTORC1 pathways (*Calcagnì et al., 2016*). This is because the fusion products maintain the DNA binding bHLH-LZ zipper domains of TFE3 and TFEB, but are under the transcriptional control of new, stronger promoters (*Kauffman et al., 2014*). Alternatively, these fusion products may act as loss of function mutations or may otherwise possess some novel transforming activity owing to their altered conformations. One TFE3 fusion, PSF-TFE3, has been reported to sequester both wild type TFE3 and p53 out of the nucleus, resulting in loss of both TFE3 and p53 transcriptional activity (*Mathur et al., 2003*). We attempted to detect an interaction between TFE3 and p53 using various immunoprecipitation protocols but were unable to do so with endogenous proteins (data not shown).

Our data suggest that TFEB and TFE3 might have a more direct role in cell cycle regulation than previously anticipated. It has been reported that TFE3 over-expression allows cells to escape from Rb-induced cell cycle arrest (*Nijman et al., 2006*). In addition, depletion of TFEB reduces proliferation of pancreatic and prostate cancer cells (*Blessing et al., 2017*; *Perera et al., 2015*), further suggesting that these transcription factors function as oncogenes. In agreement with these studies, we found that the Rb and DREAM pathways remain hyperactive in TFEB/TFE3-depleted cells, resulting in reduced expression of multiple genes implicated in cell cycle progression. Importantly, our observations that TFEB and TFE3 directly regulate expression of CDK4 and CDK7 provides mechanistic insight to understand how over-expression of these transcription factors may result in cell cycle progression and abnormal proliferation.

At the same time, one can envision situations in which TFEB and TFE3 act as antioncogenes. For example, deficient TFEB and TFE3 activation in response to genotoxic stress may lead to an abnormal and inefficient p53 response, facilitating the accumulation of damaged cells. In addition, the inability to induce p53-dependent apoptosis and the sustained cell cycle arrest in response to

prolonged stress may facilitate the induction of senescence, a condition that has been linked to metastasis and therapy resistance due to the ability of senescence cells to secrete interleukins and matrix metalloproteinases (*Gonzalez-Meljem et al., 2018*). Therefore, we propose that TFEB and TFE3 are important contributors to the fine-tune balance between cell cycle arrest/repair/survival, apoptosis, and senescence/quiescence pathways.

In summary, our data suggest a model in which TFEB/TFE3 and p53 mutually reinforce each other's transcriptional signaling functions and offers further evidence that disrupting TFEB and TFE3 can result in diminished p53 signaling capacity, with wide-ranging implications for renal cell carcinomas and other cancers. Future studies investigating the p53-dependent transcriptome in renal cell carcinomas caused by TFEB and TFE3 gene fusions may yield important insights for future therapeutic interventions.

# Materials and methods

**Key resources table**

| Reagent type (species) or resource | Designation | Source or reference | Identifiers | Additional information |
|---|---|---|---|---|
| Cell line (Mus musculus) | MEF | ATCC | Cat. #: CRL-2977 | |
| Cell line (Mus musculus) | MEF p53-/- | MEF p53-/- | | David J. Kwiatkowski (Brigham and Women's Hospital) |
| Cell line (Mus musculus) | MEF TFEB/ TFE3 DKO | PMID: 26813791 | | |
| Cell line (Mus musculus) | Raw264.7 TFEB/TFE3 DKO | PMID: 27171064 | | |
| Cell line (Homo sapiens, \|human) | HeLa | ATCC | Cat. #: CCL-2 RRID: CVCL_0030 | |
| Cell line (Homo sapiens, human) | HeLa (CF7) TFEB | PMID: 19556463 | | Andrea Ballabio (Baylor College of Medicine) |
| Cell line (Homo sapiens, \|human) | ARPE19 | ATCC | Cat. #: CRL-2302 RRID: CVCL_0145 | |
| Adenovirus | TFEB (WT, S211A) | Welgen | | PMID: 24448649 |
| Adenovirus | TFE3 S321A | Welgen | | PMID: 23401004 |
| Transfected construct | pRK5-HA-GST-RagBGTP | Addgene | plasmid 19303 | PMID: 18497260 |
| Transfected construct | pRK5-HA-GST-RagDGDP | Addgene | plasmid 19308 | PMID: 18497261 |
| Transfected construct | pcDNA3.1-TFEB | Invitrogen | | PMID: 19556463 |
| Antibody | Rabbit anti-TFE3 | Sigma-Aldrich | Cat. #: HPA023881 RRID: AB_1857931 | WB (1:1000) |
| Antibody | Rabbit anti-phospho S321 TFE3 | YenZym Antibodies | | WB (1:1000) |
| Antibody | Rabbit anti-TFEB | Bethyl Laboratories | Cat. #: A303-673A RRID: AB_11204751 | WB (1:5000) |
| Antibody | Rabbit anti-TFEB | Cell Signaling Technology | Cat. #: 4240 RRID: AB_11220225 | WB (1:1000) |
| Antibody | Rabbit anti-phospho S211 TFEB | YenZym Antibodies | | WB (1:1000) |

*Continued on next page*

*Continued*

| Reagent type (species) or resource | Designation | Source or reference | Identifiers | Additional information |
|---|---|---|---|---|
| Antibody | Mouse anti-Flag | Sigma-Aldrich | Cat. #: F3165 clone M2 RRID: AB_259529 | WB (1:5000) IF (1:1000) |
| Antibody | Mouse anti-actin | BD Transduction Laboratories | Cat. #: 612656 RRID: AB_2289199 | WB (1:1000) |
| Antibody | Mouse anti-HA | Covance | Cat. #: MMS-101P RRID: AB_2314672 | IF (1:3000) |
| Antibody | Rat anti-LAMP1 | DSHB | Cat. #: 1D4B RRID: AB_2134500 | WB (1:1000) |
| Antibody | Rabbit anti-Histone H3 | Cell Signaling Technology | Cat. #: 9003 | WB (1:1000) |
| Antibody | Rabbit anti- p70 S6 Kinase | Cell Signaling Technology | Cat. #: 2708 RRID: AB_390722 | WB (1:1000) |
| Antibody | Rabbit anti-phospho-p70 S6 Kinase | Cell Signaling Technology | Cat. #: 9205 RRID: AB_330944 | WB (1:1000) |
| Antibody | Rabbit anti-4E-BP1 | Cell Signaling Technology | Cat. #: 9644 RRID: AB_2097841 | WB (1:1000) |
| Antibody | Rabbit anti-phospho-4E-BP1 | Cell Signaling Technology | Cat. #: 2855 RRID: AB_560835 | WB (1:1000) |
| Antibody | Rabbit anti-p53 | Cell Signaling Technology | Cat. #: 32532 | WB (1:1000) |
| Antibody | Mouse anti-p53 | Cell Signaling Technology | Cat. #: 2524 | WB (1:1000) |
| Antibody | Rabbit anti-acetyl-Lys379-p53 | Cell Signaling Technology | Cat. #: 2570 RRID: AB_823591 | WB (1:1000) |
| Antibody | Rabbit anti-phospho-Ser392-p53 | Cell Signaling Technology | Cat. #: 9281 RRID: AB_331462 | WB (1:1000) |
| Antibody | Rabbit anti-phospho-Ser15-p53 | Cell Signaling Technology | Cat. #: 9284 RRID: AB_331464 | WB (1:1000) |
| Antibody | Rabbit anti-Mdm2 | R and D Systems | Cat. #: AF1244 RRID: AB_2143538 | WB (1:800) |
| Antibody | Rabbit anti-Galectin-1 | Cell Signaling Technology | Cat. #: 12936 | WB (1:1000) |
| Antibody | Rabbit anti-Caspase-3 | Cell Signaling Technology | Cat. #: 9662 RRID: AB_331439 | WB (1:1000) |
| Antibody | Rabbit anti-CDK4 | Cell Signaling Technology | Cat. #: 12790 RRID: AB_2631166 | WB (1:1000) |
| Antibody | Rabbit anti-CDK7 | Cell Signaling Technology | Cat. #: 2090 RRID: AB_2077140 | WB (1:1000) |
| Antibody | Rabbit anti-phospho-Ser639-RBL2 | Thermo Fisher Scientific | Cat. #: PA564769 RRID: AB_2662148 | WB (1:1000) |
| Antibody | Rabbit anti-RBL2 | Cell Signaling Technology | Cat. #: 13610 | WB (1:1000) |
| Antibody | Rabbit anti-phospho-Ser807/811-Rb | Cell Signaling Technology | Cat. #: 8516 RRID: AB_11178658 | WB (1:1000) |
| Antibody | Rabbit anti-Rb | Cell Signaling Technology | Cat. #: 9313 RRID: AB_1904119 | WB (1:1000) |
| Antibody | Rabbit anti-phospho-Ser345-Chk1 | Cell Signaling Technology | Cat. #: 2348 RRID: AB_331212 | WB (1:1000) |
| Antibody | Mouse anti-Chk1 | Cell Signaling Technology | Cat. #: 2360 RRID: AB_2080320 | WB (1:1000) |

*Continued*

| Reagent type (species) or resource | Designation | Source or reference | Identifiers | Additional information |
|---|---|---|---|---|
| Antibody | Rabbit anti-phospho-Tyr15-cdc2 | Cell Signaling Technology | Cat. #: 4539 RRID: AB_560953 | WB (1:1000) |
| Antibody | Rabbit anti-cdc2 | Cell Signaling Technology | Cat. #: 77055 RRID: AB_2716331 | WB (1:1000) |
| Antibody | Rabbit anti-phoshpo-Thr199-NPM | Cell Signaling Technology | Cat. #: 3541 RRID: AB_331497 | WB (1:1000) |
| Antibody | Rabbit-anti-NPM | Cell Signaling Technology | Cat. #: 3542 RRID: AB_2155178 | WB (1:1000) |
| Antibody | Mouse anti-GAPDH | Santa Cruz Biotechnology | Cat. #: sc-365062 RRID: AB_10847862 | WB (1:1000) |
| Antibody | HRP conjugated donkey anti-rabbit | Cell Signaling Technology | Cat. #: 7074 RRID: AB_2099233 | WB (1:5000) |
| Antibody | HRP conjugated donkey anti-mouse | Cell Signaling Technology | Cat. #: 7076 RRID: AB_330924 | WB (1:5000) |
| Antibody | HRP conjugated goat anti-rat | Cell Signaling Technology | Cat. #: 7077 RRID: AB_10694715 | WB (1:5000) |
| Antibody | Alexa Fluor 488 donkey anti-mouse | Life Technologies | Cat. #: A-21202 RRID: AB_141607 | IF (1:5000) |
| Antibody | Alexa Fluor 488 goat anti-rat | Life Technologies | Cat. #: A-11006 RRID: AB_2534074 | IF (1:5000) |
| Antibody | Alexa Fluor 594 donkey anti-rabbit | Life Technologies | Cat. #: A-21207 RRID: AB_141637 | IF (1:5000) |
| Antibody | Alexa Fluor 594 donkey anti-mouse | Life Technologies | Cat. #: A-21203 RRID: AB_141633 | IF (1:5000) |
| Commercial assay or kit | Pacific Blue AnnexinV apoptosis detection kit | BioLegend | Cat. #: 640926 | |
| Chemical compound or drug | Etoposide | Cell Signaling Technology | Cat. #: 2200 | |
| Chemical compound or drug | dimethylformamide | Sigma-Aldrich | Cat. #: D4551 | |
| Chemical compound or drug | Cisplatin | Sigma-Aldrich | Cat. #: 479306 | |
| Chemical compound or drug | LLOMe | Sigma-Aldrich | Cat. #: L7393 | |
| Chemical compound or drug | Cycloheximide | Sigma-Aldrich | Cat. #: C1988 | |
| Chemical compound or drug | Nutlin-3 | R and D Systems | Cat. #: 3984 | |
| Chemical compound or drug | Earle's Balanced Salt Solution | Thermo Fisher Scientific | Cat. #: 14155063 | |

## Cell culture and treatments

Wild-type Mouse Embryonic Fibroblasts (MEFs) (CRL-2977, ATCC), p53$^{-/-}$ MEFs (David J. Kwiatkowski, Brigham and Women's Hospital, Boston), RAW264.7 cells (TIB-71, ATCC) and HeLa (CF7) cells stably expressing TFEB-FLAG (previously described in *Martina et al., 2012*) were grown in DMEM,

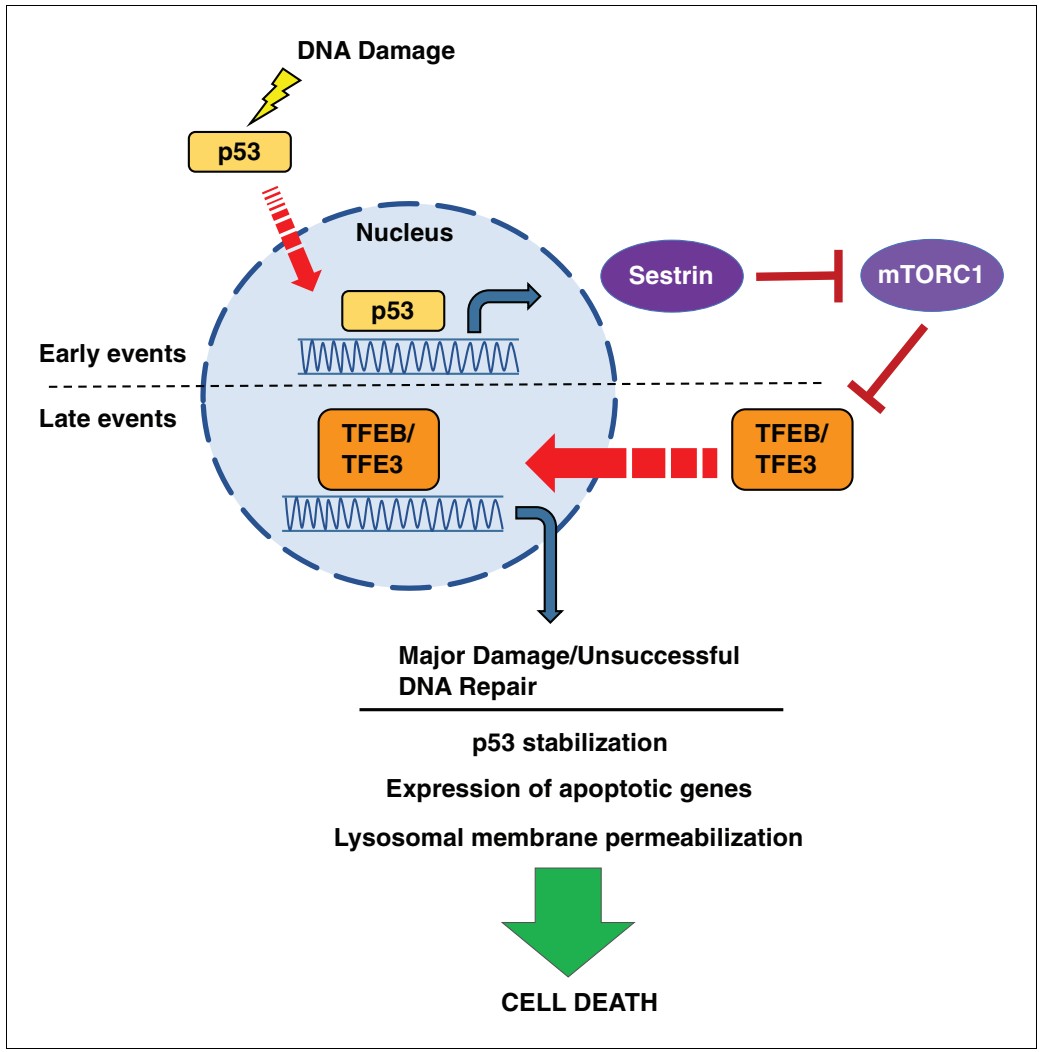

**Figure 10.** Schematic representation of a novel p53-mTORC1-TFEB/TFE3 pathway activated by DNA damage stress. Following DNA damage stress, p53 rapidly promotes the transcription of numerous downstream targets involved in DNA repair, cell cycle arrest and apoptosis. Some p53 targets, such as the members of the sestrin family, cause a reduction in mTORC1 activity, thus leading to TFEB and TFE3 activation. This p53-dependent activation of TFEB and TFE3 results in enhanced p53 signaling.
DOI: https://doi.org/10.7554/eLife.40856.038

high glucose, GlutaMAX, sodium pyruvate (Thermo Fisher 10569) supplemented with 10% fetal bovine serum (FBS). ARPE19 cells (CRL-2302, ATCC) were grown in DMEM/F-12, GlutaMAX, sodium pyruvate (Thermo Fisher 10565) supplemented with 10% FBS. TFEB and TFE3 knockout MEFs were generated as described (*Martina et al., 2016*). Cells were grown at 37°C in a humidified 5% $CO_2$ chamber. The identity of the HeLa and ARPE-19 cell lines was authenticated by short tandem repeat (STR) analysis by the supplier. The identity of the MEF and RAW264.7 lines was verified by RNA-seq analysis and functional tests (e.g. RAW264.7 cells were activated in response to LPS and live bacteria and they differentiated into osteoclasts after RANKL treatment). Cell lines were tested regularly for mycoplasma contamination and tested negative in all cases.

For drug treatments, cells were treated for the indicated times with the following reagents: DMSO (Sigma-Aldrich), 100 μM Etoposide (Cell Signaling Technology), dimethylformamide (Sigma-Aldrich D4551), 50 μM Cisplatin (Sigma-Aldrich 479306), Ethanol (Werner Graham Company), 2 mM LLOMe (Sigma-Aldrich L7393). 50 μg/ml Cycloheximide (Sigma-Aldrich C1988), 10 μM Nutlin-3 (R and D Systems 3984). For starvation, cells were washed three times in PBS and incubated in Earle's

Balanced Salt Solution (Thermo Fisher 14155063). For UV treatment cells were washed with Hank's Balanced Salt Solution (Thermo Fisher Scientific) and exposed to 30 J/cm$^2$ UV-C radiation before adding back growth media.

MEF and RAW264.7 WT and TFE3/TFEB DKO cells were generated using CRISPR-Cas9 as described previously (*Martina et al., 2016*; *Pastore et al., 2016*). Gene knockdown with siRNA and overexpression with Adenovirus infection were performed as previously described (*Martina and Puertollano, 2013*). Transfections were performed with Fugene 6 (Promega Corporation) per the manufacturer's instructions. The plasmid encoding the full-length human TFEB has been previously described (*Sardiello et al., 2009*). Amino acid substitutions in TFEB (Ser211Ala and Arg245-248Ala for NLSmut) were made using the QuickChange Lightning site-directed mutagenesis kit (Agilent Technologies) according to the manufacturer's instructions. The following constructs were obtained from Addgene: plasmid 19303, pRK5-HA-GST-RagB$_{GTP}$ and plasmid 19308, pRK5-HA-GST-RagD$_{GDP}$ (*Sancak et al., 2008*).

## Antibodies

The following antibodies were used in this study: Rabbit anti-TFE3 (Sigma HPA023881), Rabbit anti-phospho S321 TFE3 (YenZym Antibodies, produced as described in *Martina et al., 2016*), Rabbit anti-TFEB (Bethyl Laboratories, A303-673A), Rabbit anti-TFEB (Cell Signaling Technology, 4240), Rabbit anti-phospho S211 TFEB (YenZym Antibodies, produced as described in *Martina and Puertollano, 2018*), mouse anti-Flag (clone M2, Sigma-Aldrich, F3165), mouse anti-actin (BD Transduction Laboratories, 612656), mouse anti-HA (clone 16B12, Covance, MMS-101P), rat anti-LAMP1 from the Developmental Studies Hybridoma Bank deposited by August, J.T. (DSHB, 1D4B), rabbit anti-Histone H3 (Cell Signaling Technology, 9003), rabbit anti- p70 S6 Kinase (Cell Signaling Technology, 2708), rabbit anti-phospho-p70 S6 Kinase (Cell Signaling Technology, 9205), rabbit anti-4E-BP1 (Cell Signaling Technology, 9644), rabbit anti-phospho-4E-BP1 (Cell Signaling Technology, 2855), rabbit anti-p53 (Cell Signaling Technology, 32532), mouse anti-p53 (Cell Signaling Technology, 2524), rabbit anti-acetyl-Lys379-p53 (Cell Signaling Technology, 2570), rabbit anti-phospho-Ser392-p53 (Cell Signaling Technology, 9281), rabbit anti-phospho-Ser15-p53 (Cell Signaling Technology, 9284), rabbit anti-Mdm2 (R and D Systems, AF1244), rabbit anti-Galectin-1 (Cell Signaling Technology, 12936), rabbit anti-Caspase-3 (Cell Signaling Technology, 9662), rabbit anti-CDK4 (Cell signaling Technology, 12790), rabbit anti-CDK7 (Cell signaling Technology, 2090), rabbit anti-phospho-Ser639-RBL2 (Thermo fisher scientific, PA564769), rabbit anti-RBL2 (Cell signaling Technology, 13610), rabbit anti-phospho-Ser807/811-Rb (Cell signaling Technology, 8516), rabbit anti-Rb (Cell signaling Technology, 9313), rabbit anti-phospho-Ser345-Chk1 (Cell signaling Technology, 2348), mouse anti-Chk1 (Cell signaling Technology, 2360), rabbit anti-phospho-Tyr15-cdc2 (Cell signaling Technology, 4539), rabbit anti-cdc2 (Cell signaling Technology, 77055), rabbit anti-phoshpo-Thr199-NPM (Cell signaling Technology, 3541), rabbit-anti-NPM (Cell signaling Technology, 3542), mouse-anti-GAPDH (Santa Cruz Biotechnology, sc-365062), HRP conjugated donkey anti-rabbit (Cell Signaling Technology, 7074), donkey anti-mouse (Cell Signaling Technology, 7076), and goat anti-rat (Cell Signaling Technology, 7077) secondary antibodies, and Alexa Fluor 488 conjugated donkey anti-mouse (Life Technologies, A-21202), goat-anti rat (Life Technologies, A-11006), or and Alexa Fluor 594 conjugated donkey anti-rabbit (Life Technologies, A-21207) or donkey anti-mouse secondary antibodies (Life Technologies, A-21203).

## Immunofluorescence confocal microscopy

Cells grown on glass coverslips were washed with PBS and fixed with 4% formaldehyde (Electron Microscopy Sciences, 15710) diluted in PBS for 15 min at ambient temperature. Slides were washed 3 times with PBS and permeabilized for 10 min in 0.2% Triton X-100 in PBS with 10% FBS. Primary antibodies were incubated overnight at 4° C in blocking buffer (0.1% saponin (Sigma Aldrich, S4521), 0.02% sodium azide, and 10% FBS in PBS). Coverslips were washed 3 times with blocking buffer and incubated with secondary antibodies for 1 hr at ambient temperature followed by an additional three washes in blocking buffer and a final wash in PBS. Coverslips were mounted with Prolong Gold anti-fade reagent with DAPI (Life Technologies, P-36931).

For LMP assays, cells on coverslips were washed with KHM buffer (20 mM HEPES pH 7.3, 110 mM Potassium Acetate, 2 mM Magnesium Acetate) followed by a pre-permeabilization treatment

for 1 min at ambient temperature in KHM buffer containing 0.01% saponin. Coverslips were washed in KHM buffer again followed by fixation in 4% formaldehyde for 15 min at ambient temperature. Coverslips were washed an additional 3 times with PBS and submerged in chilled methanol for 10 min at −20° C. Coverslips were washed in PBS and incubated overnight at 4° C in blocking buffer. Washing and secondary antibodies were added as described above.

Images were acquired with an LSM 510 Meta confocal microscope (Zeiss, Oberkochen, Germany) with 63x numerical aperture 1.4 oil immersion objective with a Zeiss AxioCam camera. Images are presented as maximum projections through two images acquired 1 μm apart in the Z-plane.

## Subcellular fractionation

Cells were lysed in NP-40 lysis buffer (l0 mM Tris, pH 7.9, 140 mM KCl, 5 mM $MgCl_2$ and 0.5 % NP-40) supplemented with protease and phosphatase inhibitors and kept on ice for 15 min. The lysates were then centrifuged at 1000 x g for 5 min. The supernatant represents the cytosolic plus the membrane fraction. The pellets (nuclear fraction) were washed twice in NP-40 lysis buffer and then sonicated in Laemmli sample buffer.

## Western blots

Cells were washed with PBS and lysed in buffer containing 50 mM Tris, pH 7.4, 150 mM NaCl, 0.1% Triton X-100 (Sigma Aldrich, T9284), with protease inhibitor (Roche, 11836170001) and phosphatase inhibitor cocktails (Roche, 04906837001). Whole cell lysates were homogenized by passing through 25G needles and centrifuged at 16,000 rcf at 4°C. Total protein concentration measured using DC Protein Assay (Bio-Rad 500–0116). Soluble fractions were mixed with NuPage 4X loading buffer (Life Technologies, NP0007) and 10X reducing agent (Life Technologies, NP0009) and heated at 98° C for 5 min. Equal amounts of protein from each sample were run on Novex 4–20% Tris-Glycine gels (Life Technologies, EC6025) and transferred to 0.2 μm nitrocellulose membranes (GE Healthcare, 10600004). Blots were blocked for 1 hr at room temperature in TBS (Quality Biological, 351-086-101) with 0.05% Tween 20 (Sigma Aldrich, P7949) (TBS-T) and 5% nonfat milk. Primary antibodies were incubated overnight at 4° C in TBS-T with either 5% nonfat milk or BSA (Sigma Aldrich, A3294). HRP-conjugated secondary antibodies were incubated 1 hr at room temperature. Blots were washed with TBS-T, 3 times, 10 min each after both primary and secondary antibody incubations.

Blots were developed with Western Lighting Plus-ECL (Perkin-Elmer, NEL104001EA) or Clarity Western ECL Substrate (Bio-Rad, 1705060) and exposed on Biomax Light Film (Carestream Health, 876–1520) or imaged using a GE Healthcare Life Sciences Amersham Imager 600. Blots were quantitated with densitometric analysis using ImageJ (NIH) and normalized to ACTB/β-actin or GAPDH loading controls.

## RNA-Seq sample processing

WT and TFE3/TFEB DKO MEFs or Raw264.7 cells were plated on 10 cm dishes to 75% confluence and treated for 8 hr with either DMSO control or 100 μM Etoposide. RNA was isolated from samples with the PureLink RNA Mini Kit (Thermo Fisher 12183018A) or RNeasy Plus Mini Kit (QIAGEN 74134). Genomic DNA was removed from RNA preparations by DNAse digestion using QIAGEN RNAse-free DNAse Set (QIAGEN 79254) according to the manufacturer's instructions.

The sequencing libraries were constructed from 0.5 ng of total RNA using the Illumina's TruSeq Stranded Total RNA kit with Ribo-Zero following the manufacturer instruction. The fragment size of RNA-seq libraries was verified using the Agilent 2100 Bioanalyzer (Agilent) and the concentrations were determined using Qubit instrument (LifeTech). The libraries were loaded onto the Illumina HiSeq 3000 for $2 \times 75$ bp paired end read sequencing and generated about 60M reads per sample. The fastq files were generated using the bcl2fastq software for further analysis.

## RNA-Seq analysis

Rigorous quality controls of paired-end reads were assessed using FastQC tools (Babraham Bioinformatics). If required, adapter sequences and low-quality bases were trimmed using Cutadapt (*Martin, 2011*). Reads were aligned to the reference genome using the latest version of HISAT2, which sequentially aligns reads to the known transcriptome and genome using the splice-aware aligner built upon HISAT2 (*Kim et al., 2015*). A rigorous validation demonstrated this procedure

outperforms other splice-aware aligners for accurately mapping simulated spliced reads, with only a slightly lower alignment yield (*Pertea et al., 2016*; *Wang, 2018*). Only uniquely mapped paired-end reads were then used for subsequent analyses. HTSeq was used for gene level abundance estimation using the GENCODE comprehensive gene annotations (*Anders et al., 2015*; *Harrow et al., 2012*). Principal component analysis (PCA) was used to assess outlier samples.

Differential expression analysis comparing cases versus controls at the gene levels of summarization were then carried out using open source Limma R package (*Smyth, 2005*). Limma-trend, was employed to implement a gene-wise linear modelling which incorporate a global mean-variance to allow robust empirical Bayes procedure in which hyper-variable genes are identified and treated separately (*Ritchie et al., 2015*). We adjust for multiple testing by reporting the FDR q-values for each feature (*Madar and Batista, 2016*). Features with $q < 0.05$ declared as genome-wide significant.

We then used the R statistical software environment using the GAGE Bioconductor packages to carry out the analyses on pre-defined gene ontology (GO) gene sets by conducting two sample t-tests on the log-based fold changes of target gene set and control sets (*Luo et al., 2009*). Both up-Cellular Component (CC) and Molecular Function (MF). FDR q-values were estimated to correct the p-values for the multiple testing issue.

## Quantitative Real-Time PCR

RNA was isolated from samples with the PureLink RNA Mini Kit (Thermo Fisher 12183018A) and reverse transcribed using SuperScript III First-Strand Synthesis SuperMix kit (Thermo Fisher, 11752). Immunoprecipitated genomic DNA was used directly for ChIP-qPCR experiments. Quantitative real-time PCR reactions were set up in triplicate with 50 ng cDNA per reaction and 200 nM forward and reverse primers along with SYBR Green PCR Master Mix (Applied Biosystems, 4309155). Reactions were run and analyzed using a QuantStudio 12K Flex Real-Time PCR system (Applied Biosystems, Life Technologies). Relative expression levels were displayed relative to control conditions and normalized using Actb (mouse genes) or ACTB and RPS18 (human genes) using the $\Delta\Delta$CT method. ChIP-qPCR data were expressed as the percentage of the non-immunoprecipitated input sample. Primers used for experiments are provided in *Supplementary file 4*.

## Cell viability and apoptosis assays

RAW WT and TFE3/B DKO cells were plated at $7.0 \times 10^6$ cells per 10 cm dishes so that they reached 75% confluence at the time of treatment (etoposide treated cells). After DMSO control treatment and 16 to 24 hr etoposide treatment, live and dead cells were collected by combining fractions of cells dissociated with Cellstripper Buffer (Corning, 25–056 CI) and those recovered suspended in the growth media. Fractions were spun at 250 rcf at 4° C and washed twice in FACS buffer (Hank's Balanced Salt Solution, 1% Bovine Serum Albumin). Cells counted and re-suspended in Annexin V binding buffer (Pacific Blue Annexin V Apoptosis Detection Kit with 7-AAD, BioLegend) at a concentration of $1.0 \times 10^7$ cells/ml. 100 µL aliquots were distributed along with a control for each sample with no dye added. Each sample received 5 µL PacificBlue-labelled Annexin V and 5 µL 7-AAD and gently mixed. Samples were incubated in the dark for 15 min at ambient temperature. Samples were washed twice with 5 mL FACS buffer and re-suspended in 400 µL binding buffer.

Flow cytometry was performed using a BD LSRII with $5.0 \times 10^5$ counts per condition. After gating, cells were defined and quantitated as viable (Annexin V-/7-AAD-), early apoptotic (Annexin V+/7-AAD-), late apoptotic (Annexin V+/7-AAD+), or necrotic (Annexin V-/7-AAD+).

## Statistical analysis

Obtained data were processed in Excel (Microsoft Corporation) and Prism (GraphPad Software) to generate bar charts and perform statistical analyses. Student's t test or Two-way ANOVA and pairwise post-tests were run for each dependent variable, as specified in each figure legend. All data are presented as mean ± SD. $p \leq 0.05$ was considered statistically significant (*) and $p \leq 0.001$ extremely significant (***). $p > 0.05$ was considered not significant (ns).

## Acknowledgements

This project was supported by the Intramural Research Program of the NIH, National Heart, Lung, and Blood Institute (NHLBI). Special thanks to J Philip McCoy and Pradeep Dagur in the NHLBI Flow

Cytometry Core and to Yuesheng Li, Yan Luo and Poching Liu in the NHLBI DNA Sequencing and Genomics Core. Thanks also to Kenneth Kraemer and Sikander Khan for their assistance with UV exposure experiments.

## Additional information

### Funding

| Funder | Author |
| --- | --- |
| National Institutes of Health | Eutteum Jeong<br>Owen A Brady<br>Jose A Martina<br>Mehdi Pirooznia<br>Iker Tunc<br>Rosa Puertollano |

The funders had no role in study design, data collection and interpretation, or the decision to submit the work for publication.

### Author contributions

Owen A Brady, Conceptualization, Data curation, Formal analysis, Writing—original draft; Eutteum Jeong, Conceptualization, Data curation, Formal analysis, Writing—review and editing; José A Martina, Data curation, Formal analysis, Writing—review and editing; Mehdi Pirooznia, Ilker Tunc, Data curation, Formal analysis; Rosa Puertollano, Conceptualization, Formal analysis, Supervision, Funding acquisition, Investigation, Project administration, Writing—review and editing

### Author ORCIDs

Rosa Puertollano (iD) http://orcid.org/0000-0002-1106-5489

### Decision letter and Author response

Decision letter https://doi.org/10.7554/eLife.40856.047
Author response https://doi.org/10.7554/eLife.40856.048

## Additional files

### Supplementary files

• Supplementary file 1. RNA-Seq data displaying differential gene expression from WT versus TFE3/TFEB DKO MEFs exposed to 100 μM etoposide for 8 hr.
DOI: https://doi.org/10.7554/eLife.40856.039

• Supplementary file 2. RNA-Seq data displaying differential gene expression from WT versus TFE3/TFEB DKO RAW264. 7 cells exposed to 100 μM etoposide for 8 hr.
DOI: https://doi.org/10.7554/eLife.40856.040

• Supplementary file 3. Expression of genes regulated by the p53-DREAM pathway in WT and TFEB/TFE3 DKO RAW264. 7 cells under control and etoposide-treated conditions.
DOI: https://doi.org/10.7554/eLife.40856.041

• Supplementary file 4. List of all primers used in this study.
DOI: https://doi.org/10.7554/eLife.40856.042

• Transparent reporting form
DOI: https://doi.org/10.7554/eLife.40856.043

### Data availability

RNA-seq data has been deposited in GEO under accession number GSE118518. The Metadata sheets have been included as supplementary files

The following dataset was generated:

**Database and**

| Author(s) | Year | Dataset title | Dataset URL | Identifier |
|---|---|---|---|---|
| Brady OA, Jeong E, Martina JA, Pirooznia M, Tunc I, Puertollano R | 2018 | DNA Damage Response in control and TFEB/TFE3 double knockout cells treated with Etoposide | https://www.ncbi.nlm.nih.gov/geo/query/acc.cgi?acc=GSE118518 | NCBI Gene Expression Omnibus, GSE118518 |

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
