## [Decision Letter]

Thank you for submitting your article "TFE3 and TFEB Amplify p53 Dependent Transcriptional Programs in Response to DNA Damage" for consideration by *eLife*. Your article has been reviewed by three peer reviewers, one of whom is a member of our Board of Reviewing Editors, and the evaluation has been overseen by Kevin Struhl as the Senior Editor. The reviewers have opted to remain anonymous.

The reviewers have discussed the reviews with one another and the Reviewing Editor has drafted this decision to help you prepare a revised submission.

Summary:

The transcription factors TFEB and TFE3 have been well characterized for their functions in regulating the autophagy-lysosome pathway. In this manuscript, Brady and colleagues reported a role of TFEB/TFE3 in DNA-damage response (DDR). They found that TFE3 and TFEB are activated by DNA damage in a p53-mediated mTORC1 inhibition dependent manner. The activated TFEB/TFE3 in turn promotes the stabilization of p53, thus amplifying p53 signaling in DDR. TFEB/TFE3 double KO (DKO) cells show alterations in the DNA damage response. In TFEB/TFE3 DKO cells, genes involved in lysosome membrane permeabilization (LMP) and apoptosis are dysregulated. Prolonged DNA damage causes impaired LMP and apoptosis induction in DKO cells. Finally, the authors demonstrated a previously unrecognized role of TFEB and TFE3 in the regulation of cell cycle checkpoints in response to DNA damage. In general, this is an interesting paper and most of the experiments are convincingly performed. However, several important questions remain to be addressed to establish the role of TFEB/TFE3 in p53-dependent transcriptional program.

Essential revisions:

1) The authors switched cell types throughout the manuscript. It is unknown whether the conclusion obtained in one cell line can be applied to other cell lines. Some experiments should be performed in multiple cell lines. For instance, in Figure 1, they should show all read-outs (IF and Western blot) with all forms of DNA damage (etoposide, doxorubicin, UV, etc.) in all cell types (RAW cells, MEFs, etc.). Expression of the genes reported in Figure 3 should also be performed in RAW264.7 cells or HeLa cells.

2) The dependence of mTORC1 inhibition for TFEB/TFE3 nuclear translocation following DNA damage should be experimentally tested. This can be done in cells in which mTORC1 is constitutively active.

3) The authors claim that TFEB/TFE3 regulate cell cycle. In light of this claim, the authors need to be careful that the cause of all the other differences they see in TFEB/TFE3 DKO cells is not due to cell cycle changes. For instance, all of the gene expression changes from their RNA-seq reflected in Figure 3 could easily be due to the cells just being in a different phase of the cell cycle. This applies throughout the manuscript. Please address this concern.

4) The authors examined immunofluorescence images and found an increase in the levels of TFE3 in the nucleus relative to the cytosol in response to genotoxic stress. Cellular fractionation assays should be performed to determine protein levels of nuclear-localized TFE3 and TFEB in response to DNA damage.

---

## [Author Response]

Essential revisions:1) The authors switched cell types throughout the manuscript. It is unknown whether the conclusion obtained in one cell line can be applied to other cell lines. Some experiments should be performed in multiple cell lines. For instance, in Figure 1, they should show all read-outs (IF and Western blot) with all forms of DNA damage (etoposide, doxorubicin, UV, etc.) in all cell types (RAW cells, MEFs, etc.).

As requested, we now show immunofluorescence and Western-blot analysis of four different cell types (MEFs, HeLa, ARPE-19 and RAW 264.7) treated with several DNA damage agents. These results are now shown in Figure 1 and Figure 1—figure supplement 1A-G in the revised version of the manuscript.

Expression of the genes reported in Figure 3 should also be performed in RAW264.7 cells or HeLa cells.

Please note that the RNA-seq analysis performed in RAW264.7 cells (Supplementary file 2) confirmed that most of the genes reported in Figure 3 were also significantly downregulated in TFEB/TFE3 DKO RAW264.7 cells. This included Rad9a, Chek2, Trp53inp1, Mdm2, Bbc3, Sesn1, Sesn2, Dram1, Cdkn1a, Laptm5, Ctsd, Wrap53, and Foxo3. To further corroborate these results, we have now performed qRT-PCR in untreated and etoposide treated WT and TFEB/TFE3 DKO RAW264.7 cells, confirming reduced expression of multiple p53 targets in TFEB/TFE3 DKO RAW264.7 cells (Figure 3—figure supplement 1).

2) The dependence of mTORC1 inhibition for TFEB/TFE3 nuclear translocation following DNA damage should be experimentally tested. This can be done in cells in which mTORC1 is constitutively active.

To further confirm the role of mTORC1 in TFEB/TFE3 activation in response to genotoxic stress, we assessed whether constitutive mTORC1 activation prevents nuclear translocation of these transcription factors under DNA damage conditions. For this, we transfected ARPE-19 cells with a constitutive active version of Rag GTPases, in which RagB and RagD are locked in their GTP-bound and GDP-bound states, respectively, thus becoming insensitive to mTORC1 inactivation by the p53-sestrin-GATOR1 axis. As expected, expression of active Rags, and consequent constitutive mTORC1 activation, prevented translocation of endogenous TFE3 to the nucleus following etoposide treatment. These data corroborate that mTORC1 inactivation is required for TFE3 activation in response to DNA damage. These results are now shown in Figure 2G and 2H.

3) The authors claim that TFEB/TFE3 regulate cell cycle. In light of this claim, the authors need to be careful that the cause of all the other differences they see in TFEB/TFE3 DKO cells is not due to cell cycle changes. For instance, all of the gene expression changes from their RNA-seq reflected in Figure 3 could easily be due to the cells just being in a different phase of the cell cycle. This applies throughout the manuscript. Please address this concern.

In order to address the reviewers’ concern, we compared the proliferation rate of WT and TFEB/TFE3-depleted cells in stationary state. For this we used the Click-iT Plus EdU Proliferation Kit (Thermo Fisher) and analyzed the cells by flow cytometry. As seen in Author response image 1, we did not observe differences in cell cycle progression between WT and TFEB/TFE3 DKO cells (Author response image 1). Quantification of three independent experiments revealed that similar percentages of cells were found in G0/G1, S, and G_2_/M phase in both cell types (Author response image 1). Therefore, these results suggest that the different behavior of TFEB/TFE3 DKO cells in response to DNA damage is not due to cell cycle alterations.

**Author response image 1. respfig1:** Analysis of cell cycle in WT and TFEB/TFE3 DKO RAW 264.7 cells. A.WT and TFE3/TFEB DKO Raw264.7 cells were treated with 10 μM EdU for 2 hours, DNA contents were detected with Alexa Flour 488 fluorescence by flow cytometry according to the recommended staining protocol. B.Data taken from three independent experiments and significance tested using Student’s t-test.

In addition, please keep in mind that in Figure 8 of the manuscript, we compared expression of over 250 genes directly implicated in cell cycle regulation in WT and DKO cells. Note that there are not significant differences in the expression of cell cycle genes between the two cell types in untreated cells, again arguing against “constitutive” alterations in cell cycle regulation in DKOs. It is only when we treat cells with etoposide for 8h that we detect significant differences between WT and TFEB/TFE3-depleted cells (Figure 8).

4) The authors examined immunofluorescence images and found an increase in the levels of TFE3 in the nucleus relative to the cytosol in response to genotoxic stress. Cellular fractionation assays should be performed to determine protein levels of nuclear-localized TFE3 and TFEB in response to DNA damage.

As requested by the reviewers, we have now performed subcellular fractionation in MEFs treated with either etoposide or cisplatin. As seen in Figure 1D and Figure 1—figure supplement 1H, we observed a very clear increase in the amount of nuclear TFEB and TFE3 following genotoxic stress.